# Effects of Generative Chatbots in Higher Education

**Galina Ilieva** [1,*] **, Tania Yankova** [1] **, Stanislava Klisarova-Belcheva** [1] **, Angel Dimitrov** [1] **, Marin Bratkov** [2] **and Delian Angelov** [3]

1  Department of Management and Quantitative Methods in Economics, University of Plovdiv Paisii Hilendarski, 4000 Plovdiv, Bulgaria
2  Department of Marketing and International Economic Relations, University of Plovdiv Paisii Hilendarski, 4000 Plovdiv, Bulgaria
3  Department of Economic Sciences, University of Plovdiv Paisii Hilendarski, 4000 Plovdiv, Bulgaria
*  Correspondence: galili@uni-plovdiv.bg

**Abstract:** Learning technologies often do not meet the university requirements for learner engagement via interactivity and real-time feedback. In addition to the challenge of providing personalized learning experiences for students, these technologies can increase the workload of instructors due to the maintenance and updates required to keep the courses up-to-date. Intelligent chatbots based on generative artificial intelligence (AI) technology can help overcome these disadvantages by transforming pedagogical activities and guiding both students and instructors interactively. In this study, we explore and compare the main characteristics of existing educational chatbots. Then, we propose a new theoretical framework for blended learning with intelligent chatbots integration enabling students to interact online and instructors to create and manage their courses using generative AI tools. The advantages of the proposed framework are as follows: (1) it provides a comprehensive understanding of the transformative potential of AI chatbots in education and facilitates their effective implementation; (2) it offers a holistic methodology to enhance the overall educational experience; and (3) it unifies the applications of intelligent chatbots in teaching–learning activities within universities.

**Keywords:** active learning; online learning; students' engagement; generative artificial intelligence; conversational chatbots; machine learning

## 1. Introduction

The digitization of education and training has undergone a radical transformation since the initial implementation of computer technologies in corporate training programs during the mid-20th century. In the first stage of this evolution, computer-based instructions enabled employees to acquire new knowledge and skills in a self-paced and interactive manner [1]. The next stage of ICT-based learning—online learning—ensured more flexible and accessible educational opportunities, enhancing student collaboration and fostering lifelong learning. The COVID-19 pandemic with the closure of educational institutions and the shift to remote learning has further facilitated the adaptation of educators and students to new modes of online instruction, assessment, and grading [2]. In recent years, the demand for flexible and convenient learning options has driven the adoption of next-generation innovative educational tools—chatbots [3,4].

Chatbots (dialog agents, virtual assistants, automated chatbots, conversational agents) are a type of software applications initially designed to provide interaction between businesses and organizations and their customers typically through text or voice user interfaces [5]. They offer a more efficient and convenient way to handle routine and repetitive tasks, such as answering frequently asked questions, delivering basic customer support, and processing simple transactions. Nowadays, chatbots can be programmed to recognize

specific keywords or phrases, and use context and machine learning to understand intents, and generate relevant and personalized responses for users [6–8].

According to Market Research Future, a market analysis company, the conversational artificial intelligence (AI) market, including conversational chatbots, is anticipated to experience significant growth in the next seven years. The market size is projected to reach USD 32.5 billion with a predicted CAGR of 22.6% during the forecast period from 2022 to 2030 [9]. This forecast aligns with that of Markets and Markets, a marketing research company, which predicts a substantial growth in the next five years. The market size is projected to increase from USD 10.7 billion in 2023 to USD 29.8 billion by 2028, with a predicted CAGR of 22.6% during the forecast period [10].

Recent advancements in Large Language Models (LLMs) technology have given rise to new generations of intelligent chatbots capable of understanding user intentions, enabling natural speech-like conversation and providing interactive, personalized, and affordable guidance and support to the user [11,12]. In higher education, these chatbots answer students' questions, provide feedback and assessments, and facilitate communication and collaboration among students and instructors. The benefits for instructors are numerous as well as smart chatbots can track students' attendance, prepare presentations and tests, send assignments, and score papers and exams, reducing the time spent on routine tasks [13].

The aim of this study is to assess the influence of intelligent chatbots on students' learning in Plovdiv University Paisii Hilendarski, one of the largest academic institution in Bulgaria. To achieve this research goal, several experiments were conducted during the spring semester of the 2022–2023 academic year among students and instructors at Plovdiv University. The analysis of the collected data reveals important dependencies in students' attitudes and educators' perceptions toward AI chatbots allowing for a comparison of the obtained results with those from other universities and monitoring dynamic changes over time.

The main tasks of the research are as follows:

- Developing a new conceptual framework that facilitates a systematic approach for application of intelligent chatbots in university teaching and learning;
- Gathering a student dataset for their experience with and without chatbot support, including students' learning characteristics, perceptions, attitudes toward educational chatbots, and specific problems;
- Uncovering hidden relationships in the student data through the proposed methodology;
- Clarifying the difficulties, expectations, and benefits of using chatbots in formal and informal learning environments;
- Assessing the competence of educational chatbots in handling university learning tasks;
- Providing measures for chatbot applications and recommendations for participants in the learning process to improve the university educational practice.

The main contribution of this paper is the development of a new conceptual framework for implementing intelligent chatbots in the teaching and learning activities of university students and faculty members. This reference framework allows for the systematic assessment of students' and instructors' perceptions and readiness for chatbot-based learning in electronic environment. The impact of chatbots is evaluated using classical and intelligent methods for data analysis. Early detection of problems due to improper chatbot use could not only save costs and time, but also prevent negative consequences such as misconceptions, misunderstandings, and mistakes along with further social and economic implications for higher education institutions.

The remainder of this paper is organized as follows: Section 2 outlines the primary features of popular chatbot models, platforms, and systems, especially those designed for educational purposes. In Section 3, we summarize the results obtained from previous research on the implementation of chatbots in e-learning context. Section 4 introduces a new integrated framework for incorporating smart chatbots in pedagogical activities. This section also presents and evaluates the datasets gathered after the utilization of conversa-

tional chatbots in the learning process and compares the obtained results with those from previous studies. Finally, the paper concludes and highlights future research plans.

## 2. State-of-the-Art Review of Intelligent Chatbot Models, Platforms, and Systems

Chatbots have evolved significantly over the years, with early versions being simple rule-based systems that could only respond to a limited set of commands. As Natural Language Processing (NLP) technology improved, chatbots became more sophisticated, allowing for more complex interactions and greater customization. As generative AI and new generation chatbots are still relatively new and rapidly evolving technology, managers of companies, heads of public authorities, and other stakeholders are often not fully aware of their capabilities in service automation.

In this section, we briefly describe the main characteristics of state-of-the-art language models for smart chatbots. Then, we compare key features of the most widely used AI platforms and systems for intelligent chatbots. Finally, we examine the capabilities of generative chatbots designed for educational purposes as personalized learning assistants for students and teaching assistants for instructors.

### 2.1. Large Language Models for NLP and Their Comparison

In order to create new, unique content and simulate human conversation, intelligent chatbots utilize LLMs—large foundational models for NLP. These types of AI models combine deep learning and machine learning algorithms and can understand and/or generate natural language patterns. LLMs are trained on large-scale datasets to process user inputs and produce coherent and contextually appropriate responses for various NLP tasks. As each model possesses different advantages and disadvantages, selecting an appropriate LLM depends on a number of criteria, such as the specific requirements of NPL tasks and the characteristics of input data. The following section provides a concise overview of the most commonly employed LLMs, presented in chronological order.

BERT—Bidirectional Encoder Representations from Transformers (Google, Mountain View, CA, USA, https://cloud.google.com/ai-platform/training/docs/algorithms/bert-start, accessed on 31 May 2023) (2018) is a family of open-source transformer-based LLMs [14]. The bidirectional design of BERT allows the algorithm to consider context from both the left and right sides of each word, leading to good results on many NLP tasks, including question answering and named entity recognition [15]. The larger version of BERT is an industrial-level multilingual model with 340 million parameters. It has been trained on over 3.3 billion words (BookCorpus and English Wikipedia). Since 2019, Google has used BERT to better understand user searches. BERT can also be employed in a wide variety of language tasks such as text generation, question answering with polysemy resolution, text prediction, summarization, and sentiment analysis [16]. Several BERT models are available with improved performance for specific NLP tasks such as Multilingual BERT, domain specific BERT versions, and smaller and faster ALBERT and DistilBERT.

XLNet—Generalized autoregressive pre-training for language understanding (Carnegie Mellon University (CMU), Pittsburgh, PA, USA; Google Brain, Mountain View, CA, USA, https://github.com/zihangdai/xlnet, accessed on 31 May 2023) (2019) is an open large bidirectional language model that employs Transformer-XL method for a variety of language tasks involving long context. Unlike BERT and other LLMs, XLNet uses a permutation-based training approach that determines the joint probability distribution of all possible permutations of the input sequence [17]. This approach enables XLNet to outperform other language models on a wide range of NLP tasks, including language translation and content generation. XLNet (large version) is an English language model with 340 million parameters and 1.3 GB in size.

ERNIE—Enhanced Representation through kNowledge IntEgration) (Baidu, Beijing, China; Peng Cheng Laboratory, Shenzhen, China, http://research.baidu.com/Blog/index-view?id=183, accessed on 31 May 2023) (2019) is a series of industrial-level multilingual models designed to learn text representation enhanced by knowledge-masking strategies

(entity-level masking and phrase-level masking) [18]. As of 2019, Baidu uses ERNIE to better understand user searches. The last model version—ERNIE 3.0 Titan—has approximately 260 billion parameters. It has been trained on an over 4 TB Chinese corpus database. Titan can be fine-tuned for both Natural Language Understanding (NLU) and Natural Language Generation (NLG) tasks, such as sentiment analysis, text classification, named-entity recognition, and document retrieval and text summarization and closed-book question answering, machine translation, and dialogue generation, respectively [19].

GPT-3—Generative Pre-trained Transformer 3 (Open AI, San Francisco, CA, USA, https://platform.openai.com/docs/models/gpt-3, accessed on 31 May 2023) (2020) is a set of transformer-based models trained on extensive text datasets to generate natural-sounding human-like responses. The models have demonstrated high performance in various language-related tasks, such as language translation, question answering, and text generation [20,21]. GPT-3 has 175 billion parameters. It has been trained on approximately 500 B tokens. However, the training process is resource expensive, requiring a substantial number of machines and GPUs. To further enhance the capabilities of GPT-3, OpenAI introduced the GPT-3.5 series and GPT-3.5-Turbo model in 2023. These advancements resulted in improved performance and increased capabilities. For instance, the text-davinci-003 version can handle more complex commands, produce higher quality, and generate longer-form writing compared to earlier GPT-3 systems.

PanGu—Large-scale Autoregressive Pre-trained Chinese Language Models (Huawei, Shenzhen, China, http://www.pangu-drug.com, accessed on 31 May 2023) (2021) comprises a series of production-level large models including very large-scale transformer-based autoregressive language model [22]. These models find applications in various text generation scenarios and can tackle diverse NLP tasks (Table 1). The latest model version—PanGu-Σ—employs "sparse" (Mixture of Experts—MoE) architecture, which reduces the computing power required for training, despite having 1.1 trillion parameters. It has been trained on 40 TB of Chinese text data [23].

Wu Dao—Chinese Pre-trained Language (CPL) Model (Beijing Academy of Artificial Intelligence—BAAI, Beijing, China, accessed on 31 May 2023) (2021) is super large-scale Chinese corpora for pre-training language models [24]. BAAI later updated their product to Wu Dao 2.0 [25], which represents a pre-trained multimodal and multitasking deep learning model with MoE architecture. The new version of Wu Dao has 1.75 trillion parameters and offers support for both English and Chinese. Its capabilities span natural language processing, image recognition, and the generation of both text and images.

LaMDA—Language Model for Dialogue Application (Google, Mountain View, CA, USA, https://blog.google/technology/ai/lamda/, accessed on 31 May 2023) (2021) is a family of transformer-based neural language models designed to serve as foundational technology for dialogue-based applications [26]. Its primary purpose is to facilitate the creation of natural-sounding human language in these applications. LaMDA-based conversational interfaces can generate highly responsive and contextually relevant answers, leading to more engaging and authentic interactions between users and machines. LaMDA 2, currently in beta testing phase, possesses 137 billion parameters and was trained on approximately 1.56 TB words from publicly available dialogue data and Internet documents with over 90% of the pre-training dataset in English.

YaLM—Yet another Language Model (Yandex, Moscow, Russia, https://github.com/yandex/YaLM-100B, accessed on 31 May 2023) (2021) is a family of open-source neural networks for the production and processing of texts in both Russian and English. With its 100 billion parameters, YaLM-100B surpasses existing Russian language models in scale. The model was trained on a corpus of 1.7 TB texts from open datasets in Russian and English, enabling it to solve a variety of NLP tasks. Language models from the YaLM family comprehend the principles of text construction and can generate new texts based on linguistics rules and their knowledge of the world [27].

**Table 1.** Comparison between the most widely used LLMs.

| Name | Functionality | Supported Platforms | Access Type |
|---|---|---|---|
| BERT | Question answering, text summarization, understanding user search intentions and the content indexed by the search engine | Cloud, On-premise | Source code |
| XLNet | Question answering, sentiment analysis, search for relevant information in document bases or online | Cloud, On-premise | Source code |
| ERNIE | Chinese language understanding, literary creation, business writing, mathematical calculations, multimodal output generation | Cloud, On-premise—for previous versions | Source code (for previous versions), API |
| GPT-3 | Wide range of NLP tasks, including question answering, content generation, text summarization, text classification, information extraction | Cloud, On-premise | API |
| PanGu | Wide range of NLP tasks, including natural language inference, common sense reasoning, reading comprehension, text classification | Cloud, On-premise | Source code (for previous versions), API |
| Wu Dao | Generation of text and images, natural language processing and image recognition | Cloud, On-premise—for previous version | Source code (for the previous version), API |
| LaMDA | Language translation, text summarizing, answering information-seeking questions | Cloud, On-premise—for the previous version | Source code (after approval), API |
| YaLM | Different NLP tasks, including generating and processing text | Cloud, On-premise | Source code, API |
| PaLM | Multiple difficult tasks: Language understanding and generation, reasoning, programming code generation | Cloud, On-premise—for the previous version | Source code, API (only for the last version) |
| BLOOM | Different NLP tasks, including question answering, sentiment analysis, text classification | Cloud, On-premise | Source code, API |
| GLM-130B | Different language understanding and language generation tasks | Cloud, On-premise | Source code, API |
| LLaMA | AI developers interested in a powerful large language model | Cloud, On-premise | Source code, API |
| GPT-4 | Can perform different NLP tasks, text generation, image processing and generation | Cloud | API |

PaLM—Parallel Language Model (Google, Mountain View, CA, USA, https://palm-e.github.io/, https://ai.google/discover/palm2 accessed on 31 May 2023) (2022) is an open-source family of LLMs tailored for large-scale NLG tasks. While PaLM can generate text in response to user input, its primary focus lies in processing large amounts of data, making it suitable for tasks such as machine translation and content generation, which require intensive language processing capabilities [28]. Apart from NLG, models from the PaLM family can also be applied for advanced reasoning tasks, such as programming code and mathematical calculations, classification and question answering, translation and multilingual proficiency, and natural language generation. PaLM 2 is the latest model version. It consists of 540 billion parameters and was trained on a dataset of 3.6 trillion tokens from a combination of English and multilingual datasets including high-quality web documents, books, Wikipedia, conversations, and GitHub programming code.

BLOOM—Large Open-science Open-access Multilingual Language Model (BigScience and other research teams, https://bigscience.huggingface.co/blog/bloom, accessed on 31 May 2023) (2022) is an open-access multilingual autoregressive LLM. This model is the result of a collaboration among over 1000 researchers from various institutions and countries. It leverages industrial-scale computational resources to generate coherent text in

many programming and natural languages, often indistinguishable from human-written text. Additionally, BLOOM can act as an instruction-following model, capable of performing a wide range of text-related tasks that may not have been explicitly included in its training [29]. BLOOM consists of 176 billion parameters and was trained on approximately 366 billion tokens. The model can generate text in 46 natural languages and 13 programming languages.

GLM-130B—General Language Model (Tsinghua University, Beijing, China, https://github.com/THUDM/GLM-130B, accessed on 31 May 2023) (2022) is an open bilingual (Chinese and English) large language model. Its few-shot performance surpassed the level of the previous top model GPT-3 in the Massive Multi-Task Language Understanding (MMLU) benchmark [30]. With 130 billion parameters, GLM-130B has been trained on over 400 billion text tokens as of July 2022. The model demonstrates high performance on both language understanding and language generation tasks.

LLaMA—Large Language Model Meta AI (Meta Platforms, New York City, New York, USA, https://ai.facebook.com/blog/large-language-model-llama-meta-ai/, accessed on 31 May 2023) (2023) is a relatively new LLMs family, characterized by its efficient and resource-friendly design, making it more accessible to a broader user base. Its availability under a non-commercial license enables researchers and organizations to conveniently utilize it for their respective projects [31]. The number of LLaMAs parameters varies between 7 billion and 65 billion, with the smallest model having been trained on approximately 1 T tokens.

GPT-4—Generative Pre-trained Transformer 4 (Open AI, San Francisco, CA, USA, https://openai.com/research/gpt-4, accessed on 31 May 2023) (2023) represents a multi-modal LLM capable of accepting both image and text inputs to generate text-only outputs. It powers ChatGPT Plus chatbot and can be accessed through API via waitlist. Although specific technical details about GPT-4 are not provided, it has demonstrated stability and impressive performance on standardized benchmark tests. However, concerns remain about its bias and safety issues that need to be addressed [32]. GPT-4 has been equipped with the ability to access and utilize Internet resources.

Table 1 provides a comparison of the key characteristics, including functionality, supported platforms, and access type, for the presented language models.

The evolution of language models has undergone a shift from task-specific supervised learning approaches to the transformer architecture, specifically the self-attention mechanism [14]. Transformers, as exemplified by models like BERT and GPT-4, process entire input sequences in parallel, enabling more efficient and effective modeling of long-range dependencies compared to traditional recurrent neural networks. This architecture combines unsupervised pre-training and supervised fine-tuning, overcoming some limitations of earlier approaches. Today's Large Language Models (LLMs), trained on diverse datasets, can process input data and generate output text more accurately reflecting the context and meaning of the input text.

The main advantage of the above-mentioned LLMs is their ability to generate grammatically correct and contextually relevant natural language texts. However, this capability comes with the requirement of large amounts of data and substantial computing resources. Another key feature of described LLMs in their versatility, as a single model can effectively handle various NLP tasks including question answering, token and text classification, document summarization, translation, generation of text, and in some cases, programming code or images.

Several factors differentiate LLMs, as discussed in models' descriptions. These include the size of the training dataset, model weights, and training and inference costs. Some developers have chosen to keep their LLMs less publicized and shared to gain advantages in the first two factors. However, the third factor, which encompasses the costs of training and running large LLMs, has traditionally been affordable only for the biggest IT companies.

The language models, described in Table 1, can also be classified based on other criteria, such as model features, type of machine learning model, pre-training objectives,

modality, and software licensing model. These features can be integrated in assessment systems for choosing the most appropriate conversational AI product and its components.

Model features: With advancements in NLP methods, LLM functionality continues to expand, improving existing models and even giving rise to new model families. New modules such as Attention Condensers focus on the most relevant sections of the input, reducing the computing resources required for data analysis.

Another important characteristic of LLMs is their maximum input length (token limit). This metric indicates the maximum number of tokens that can be used in the prompt. Token limit is determined by the model architecture. For most LLMs, the input size varies between 2048 and 4096 tokens, while GPT-4 has 32 K token limit (approximately 25 K words).

Type of model: LLMs can be classified into two types according to their basic capabilities for language generation: discriminative and generative. Discriminative models (BERT-style) use deep learning algorithms, focusing on the decision boundary between classes. Given input data, the goal is to predict a specific class label. BERT, XLNet, ERNIE, and PanGu are discriminative LLMs. Generative models (GPT-style) emphasize modelling the joint distribution of inputs and outputs, generating realistic output data that aligns with the training data distribution. GPT-3, Wu Dao, LamDA, YaLM, PaLM, BLOOM, GLM, LLaMA, and GPT-4 are generative LLMs.

Unsupervised pre-training objectives: Based on their approach for generating and reconstructing sequences, LLMs can be classified into three categories: encoder-only, decoder-only, and encoder–decoder language models.

Encoder-only (auto-encoding) models aim to reconstruct the original input sequence from a compressed and abstract representation of that sequence. The encoding process captures contextual information bidirectionally, making auto-encoding models unsuitable for unconditional sequence generation. However, they are useful for tasks like language understanding, feature extraction, and text representation learning. BERT is an example of an encoder-only model.

Decoder-only (autoregressive) models generate output sequences by predicting the next token given the preceding context. These models generate text sequentially, word by word, taking into account the entire history of the sequence generated so far. This approach allows for coherent and contextually appropriate generation but may be slower and less parallelizable due to the sequential nature of the generation process. GPT-3, GPT-4, PaLM, LaMDA, BLOOM, GLM, PanGu, YaLM, and LLaMa are examples of autoregressive models.

An encoder–decoder (sequence-to-sequence) model treats each task as sequence-to-sequence conversion, which can involve text-to-text or even text-to-image or image-to-text generation. Encoder–decoder models are typically used for tasks that require both content understanding and generation, such as machine translation. XLNet is an example of an encoder–decoder model.

Modality: LLMs can be categorized into two groups, based on the number of modalities they operate with unimodal and multimodal language models. Unimodal LLMs (including all above-mentioned models except ERNIE, Wu Dao 2.0, PaLM-E, and GTP-4) operate within a single modality, typically focused on processing and generating text-based data. These models mainly deal with language-related tasks such as language generation, text classification, machine translation, and sentiment analysis. Multimodal LLMs incorporate multiple modalities, such as text, images, audio, or other forms of data. These models are designed to handle tasks that involve multiple modalities (image captioning, visual question answering, video summarization, audio–visual translation). By leveraging information from different modalities, multimodal LLMs can capture richer contextual understanding and provide more comprehensive responses. ERNIE, Wu Dao 2.0, PaLM-E, and GTP-4 are examples of multimodal LLMs.

Software-licensing model: LLMs can be divided into two groups based on their software-licensing model, open-source and proprietary software. Open-source models, such as BERT, XLNet, YaLM, PaLM, BLOOM, GLM, and LLaMA, allow researchers and developers to access, modify, and contribute to the model's development, fostering col-

laboration and innovation. In contrast, closed-source LLMs are proprietary models where the source code and related resources are not publicly available and their accessibility and customization options for the developers or organization are limited. Additionally, closed-source models may execute slowly and require higher costs. Closed-source models include ERNIE, GPT-3, PanGu, Wu Dao 2.0, LaMDA, and GPT-4.

Despite the various benefits of LLMs, they have some disadvantages and limitations, as outlined below:

- LLMs can generate biased, harmful, or inaccurate content and discriminate based on the input data they are trained on and the specific applications they are used for.
- LLMs can be vulnerable to adversarial attacks, in which attackers deliberately input misleading data to manipulate the model's output.
- Some experts have raised concerns about the environmental impact of training large language models, as it can require massive amounts of computing power and energy.

In this section, we present and compare some of the most widely used large-scale language models. Existing LLMs differ in functionalities, size, number of parameters, modality capabilities, composition, and size of the training dataset, as well as the level of customization and access methods. Various LLMs are available for different natural and programming languages and serve various industries. The recent development of these models represents significant progress in the field of NLG and opens up new possibilities for creating sophisticated AI systems.

The next section provides an overview and comparison of chatbots built upon the aforementioned LLMs. It explores diverse chatbot implementations, emphasizing their practical applications and use cases.

### 2.2. Intelligent Chatbots and Their Comparison

Chatbots are computer programs capable of simulating human conversations. In the past, chatbots relied on pre-defined conversational templates, but nowadays they utilize LLMs to understand user intents and provide relevant and personalized automated responses in real-time. These advanced AI chatbots are considered generative AI systems since they can learn patterns and structures from large datasets and create new content, such as text, images, audio, or video, similar in style, format, or meaning to the original data.

Conversational chatbots share similarities with voice assistants, but they also have distinct differences. While voice assistants primarily provide information and execute tasks upon request, AI chatbots are designed to engage in interactive dialogs with users. Furthermore, intelligent chatbots have the ability to remember and recall past conversations.

In this section, some of the most commonly used platforms and systems for intelligent chatbots are introduced and compared.

IBM Watson Assistant (IBM, Armonk, NY, USA, https://www.ibm.com/products/watson-assistant, accessed on 31 May 2023) (2016) is a cloud-based conversational AI platform for building and deploying chatbots and virtual assistants. It utilizes various AI models to understand user input and provide relevant responses [33]. Watson Assistant can be trained on a wide range of topics and customized to meet the needs of individual businesses or organizations. The platform offers pre-built integrations with popular messaging software, voice assistants, and other services, as well as analytics and reporting tools to monitor chatbot performance and identify areas for improvement.

Amazon Lex (Amazon, Seattle, DC, USA, https://aws.amazon.com/lex, accessed on 31 May 2023) (2017) is a cloud-based conversational AI service that enables developers to build and deploy chatbots and other voice- and text-based conversational interfaces. The service utilizes statistical models, deep learning neural networks, and other AI techniques to analyze natural language inputs and generate suitable responses [34]. Amazon Lex's pre-built bots reduce the deployment time of conversational AI solutions.

ERNIE Bot (https://yiyan.baidu.com, accessed on 31 May 2023) (March 2023) is an AI-powered multimodal chatbot developed by Baidu. It can comprehend human intentions and deliver accurate, logical, and fluent responses, approaching a level comparable to

humans. ERNIE Bot has undergone training using Reinforcement Learning via Human Feedback (RLHF), prompt learning, retrieval augmentation, and dialogue augmentation. Currently, ERNIE Bot application is open only to trial users.

ChatGPT (https://chat.openai.com, accessed on 31 May 2023) is an intelligent chatbot based on GPT-3.5 LLM (2022) capable of simulating human-like conversations. Users can engage with ChatGPT through a web-based dialogue interface, asking questions or requesting information on various topics within the model's training scope. ChatGPT is optimized for human dialogue using RLHF. While it is useful for tasks such as copywriting, translation, search, and customer support, it should be noted that ChatGPT's information is up until September 2021 as it does not have direct access to the latest updates from the Internet. Basic (non-peak) access to ChatGPT does not require a subscription, making it suitable for educational purposes. There are many ChatGPT plugins for different use cases.

PanGu-Bot (demo version not currently available) (2022) is a Chinese dialogue model based on the large pre-trained model PanGu-$\alpha$. It has 350 M and 2.6 B parameters and achieves good open-domain dialogue performance with high training efficiency. PanGu-Bot versions demonstrate the possibility of building high-quality software with limited dialogue data, competing with state-of-the-art dialogue systems [35].

Google Bard (https://bard.google.com, accessed on 31 May 2023) is a chatbot that engages in conversational simulations with humans and leverages web-based information to deliver responses to user inquiries. Bard also possesses the ability to assist with programming and software development tasks. Google has developed a series of Bard models, including "Multi-Bard" and "Big Bard", with varying numbers of parameters.

Yandex (YaLM) Chatbot (https://ya.ru/, accessed on 31 May 2023) (2022) is available on the main page of the Yandex search engine site, ya.ru, and is also incorporated into the virtual assistant Alice. With the integration of YandexGPT (YaLM 2.0), users can now assign tasks to Alice directly from any desktop or mobile browser.

BLOOMChat (https://huggingface.co/sambanovasystems/BLOOMChat-176B-v1, accessed on 31 May 2023) (2023) is a multilingual AI chatbot based on BLOOM LLM. The chatbot has been instruction-tuned (instruction-following) with English-focused assistant-style conversation datasets. BLOOMChat is still in its early stage of exploration.

ChatGLM-6B (https://huggingface.co/THUDM/chatglm-6b, accessed on 31 May 2023) (2023) is a bilingual (Chinese and English) language model. Despite its large size, the model can be executed on consumer-grade GPUs through quantization. ChatGLM is specifically optimized for the Chinese language and shares similarities with ChatGPT. Notably, ChatGLM is one of the few LLMs obtainable under an Apache-2.0 license, enabling commercial usage.

Alpaca (Stanford University, https://crfm.stanford.edu/2023/03/13/alpaca.html, demo version: https://chat.lmsys.org/, select Alpaca-13B accessed on 31 May 2023) (2023) is an open-source project based on LLaMA LLM. Developed by a team of researchers at Stanford University, Alpaca is designed to understand and execute tasks based on user instructions. The Alpaca-7B model has been trained on a dataset of 52 K demonstrations of instruction-following. Alpaca exhibits behavior similar to text-davinci-003 on the self-instruct evaluation set but is smaller in size and more cost-effective to reproduce, making it an attractive ChatGPT alternative [36].

Vicuna (UC Berkeley; CMU; Stanford University; Mohamed bin Zayed University of Artificial Intelligence—Abu Dhabi, UAE; UC San Diego, https://vicuna.lmsys.org/, demo version: https://chat.lmsys.org, select Vicuna-7B or Vicuna-13B, accessed on 31 May 2023) (2023) is a chatbot trained through the fine-tuning of the LLaMA model using conversations contributed by users and collected from ShareGPT, a Chrome extension built upon ChatGPT. Initial evaluations indicate that the 13B parameter version of Vicuna surpasses the performance of both LLaMA and Alpaca and comes close to Bard and ChatGPT-4 [37].

ChatGPT Plus (https://openai.com/blog/chatgpt-plus, accessed on 31 May 2023) (March 2023) is a subscription-based Web service operating on GPT-4 LLM. This multimodal

platform can process not just text but also images, including photographs, diagrams, and screenshots. Compared to ChatGPT, ChatGPT Plus offers higher-quality responses due to the advanced language generation and text completion capabilities of GPT-4, as well as its significantly larger number of model parameters in comparison to GPT-3.5. Key features of ChatGPT Plus include improved response accuracy, faster response times, enhanced creative capabilities, significantly longer text inputs and outputs, and real-time access to online data.

Microsoft Bing and Edge Chat (https://www.bing.com, accessed on 31 May 2023) is a chatbot based on GPT-4 LLM. It is accessible directly from the Bing search page and integrated into the Edge sidebar. The chatbot can be applied for various NLP tasks, including writing, summarization, translation, and conversation. Similar to ChatGPT, users can type questions directly into the interface and receive responses, with the option to ask follow-up questions. Conversational AI provides up-to-date information and relevant website links. The Edge integration of the chatbot enables automatic PDF merging and the generation of LinkedIn content. Furthermore, Edge understands the web pages being visited by the user and adapts its functionality accordingly.

The AI chatbots mentioned above utilize machine learning and other advanced AI techniques to provide conversations that closely resemble human interaction. These intelligent chatbots automate routine tasks, offer personalized communication, and enable 24/7 availability. Moreover, they have the ability to collect data and uncover valuable insights that can enhance business processes and support strategic decision-making.

AI chatbots can be developed for various online platforms and messaging channels, such as websites, social media platforms, messaging software, and voice assistants. They are typically integrated with backend systems, databases, and mobile applications to access and retrieve information needed to perform their tasks. Intelligent chatbots can execute a wide range of business functions, including sales and marketing, personal assistance, and information retrieval.

Depending on their main features, the above-mentioned AI chatbots' platforms and systems can be classified based on several criteria.

- Functionality: The mentioned chatbots are specifically designed for natural language understanding and generation. The majority of them are constructed using LLMs and transformer architecture. However, there are a few exceptions such as IBM Watson Assistant and AWS Lex, which leverage a combination of various AI methods.
- Language support: ERNIE Bot supports multiple languages, including Chinese and English, while ChatGPT primarily focuses on English language tasks.
- Internet connectivity: Only ERNIE, PanGu, Bard, and ChatGPT Plus have Internet access and can receive real-time information.
- Multi-modality: Only ERNIE Bot, Bard, and ChatGPT Plus can process multimodal inputs (text or images).
- Pricing: PanGu, ChatGPT, and GPT Plus are paid software. These chatbot platforms offer flexible pricing models and even provide free plans for individual users and businesses.
- Ease of use: Some of the available chatbot alternatives, like ChatGPT and ChatGPT Plus, offer user-friendly interfaces that make them more accessible to users without coding knowledge. The ease of use varies among different chatbot options.
- Complexity of set-up: It is an important consideration when choosing a chatbot. Some chatbots may require programming expertise to set up and deploy effectively.
- Use cases: While chatbots have a wide range of applications in NLP, ChatGPT and ChatGPT Plus are particularly useful for conversational interactions, making them suitable for chatbot and virtual assistant applications. On the other hand, ERNIE Bot and BLOOMChat with their multi-language support are ideal for tasks involving multiple languages or cross-lingual applications. Google Bard demonstrates enhanced mathematical skills and improved logical reasoning capabilities. Alpaca and Vicuna

models can accelerate the progress in the development of reliable AI chatbots. These models offer capabilities comparable to closed-source models like text-davinci-003.

Despite their numerous advantages, the chatbots have some drawbacks:

1. Often, chatbots lack real-time data generation capabilities, which hinders their ability to instantly monitor customer conversations and promptly identify potential issues.
2. Chatbots can sometimes generate inaccurate or "hallucinated" responses, which necessitates extensive and time consuming fact-checking.
3. Another problem is the occasional inability of chatbots to understand users' questions and requests. This problem arises from users' lack of knowledge about structuring their questions effectively to elicit responses that meet their specific needs.

Furthermore, similar to any emerging information technology, AI chatbots can give rise to ethical and social concerns. These concerns include the potential application of subliminal or targeted manipulative techniques, exploitation of user vulnerabilities, and unregulated evaluation of users based on socio-economic characteristics. To mitigate such risks, the European Parliament has been working on the development of the AI Act for several years. The AI Act outlines guidelines and requirements for the development and deployment of AI technologies, with a strong emphasis on promoting the ethical and responsible use of AI systems. Its objective is to safeguard fundamental human rights, including privacy and non-discrimination.

Chatbots are LLM-based AI systems that can recognize complex semantic and syntactic dependencies in texts and generate diverse and well-structured responses. Moreover, their flexibility and capacity to learn from new data enable AI chatbots to tackle novel tasks and adapt to changing conditions.

### 2.3. Educational AI Chatbots

Since the late 1960s, numerous educational innovation companies have been developing chatbots to simulate conversations for language practice. In the 21st century, with the rise of conversational AI, chatbots have been embedded into traditional e-learning systems, becoming an integral part of educational experiences in schools, universities, and other educational institutions [38,39].

Educational chatbots are chatbots designed for pedagogical purposes [40]. Educational AI chatbots utilize NLP techniques to interact with students and lecturers in a conversational manner. They are programmed to provide information, deliver educational content, offer personalized learning support, and engage in interactive learning experiences [41].

The following are key characteristics of generative AI chatbots as a pedagogical tool:

- Conversational assistance: They understand and address students' questions in a conversational manner.
- Multi-modality: They can support multiple modes of communication, including text, speech, and visual elements.
- Multilingual support: They offer multilingual capabilities for diverse student communities.
- Cost-effectiveness and scalability: They are capable of handling large student populations while remaining cost-effective.
- Integration with other software systems: They can integrate with learning management systems, library databases, and online search tools.
- Data analytics and insights: They can provide instructors with data analytics and insights to enhance their teaching methods [42].

In comparison to Learning Management Systems (LMS) and Massive Open Online Courses (MOOCs), intelligent chatbots offer a more interactive and conversational learning experience that can be tailored to individual students. Unlike Intelligent Tutoring Systems (ITS), smart chatbots are not limited to specific domains and can be applied in various educational contexts. Moreover, AI chatbots have the capability to provide personalized real-time feedback and assistance to learners, similar to ITS.

The use of new generation chatbots in education is rapidly expanding due to numerous reasons. One of the most significant is their ability to process and understand natural language at an unprecedented level. In addition to their natural language processing capabilities, AI chatbots are also incredibly versatile. They can be trained to perform a wide range of tasks for assisting of teaching and learning, supporting some in-class activities and a variety of out-of-class and some in-class activities. They ensure more effective methods for interactive and effective student learning and minimize teachers' workload.

AI chatbots offer particular value in higher education due to the following reasons:

1. Advanced and specialized subjects benefit from personalized assistance;
2. Emphasis on self-directed and independent learning;
3. Access to scholarly resources, aiding student research activities such as literature reviews and research methodology guidance.

Smart chatbots can transform many teaching–learning activities from personalized tutoring and homework assistance to administrative tasks and student grading. Opening new possibilities for creativity and innovation in education, AI chatbots have the potential to reshape the educational landscape.

## 3. Related Work

### 3.1. Theoretical Frameworks for Application of AI Chatbots in Education

In this section, we review the characteristics of existing frameworks for AI chatbots-based learning in higher education institutions.

Some research is focused on distinct aspects of the applications of a particular educational chatbot. Gimpel et al. [43] have devised a practical, step-by-step guidance for students and lecturers, providing exemplary prompts on how to utilize ChatGPT for learning and teaching purposes. Furthermore, the authors have outlined the potential transformation of the complete student lifecycle, encompassing admissions, enrollment, career services, and other areas of higher education management. Rasul et al. [44] have presented a framework for usage of ChatGPT in universities based on the constructivism learning theory. In the new framework, the chatbot offers support in generating ideas, conducting research, analyzing information, and writing tasks. However, there are risks of academic misconduct, bias, false information, and inadequate assessment design. Hence, both educators and students must be careful to ensure the ethical, reliable, and effective implementation of this AI technology.

Other researchers provide comprehensive frameworks for the implementation of chatbots in education; however, they focus only on specific aspects of the learning process. Su and Yang [45] has proposed a theoretical framework to guide the utilization of ChatGPT and other generative AI in education, aiming to improve teaching and learning outcomes. This framework has been applied in the creation of virtual coaches that offer feedback to teachers. Chan [46] has created a new AI Ecological Education Policy Framework with pedagogical, ethical, and operational dimensions. This framework effectively addresses concerns related to privacy, security, and accountability while also emphasizing infrastructure and training requirements. Sajja et al. [47] have developed an AI-augmented intelligent educational assistance framework based on GPT-3 that automatically generates course specific intelligent assistants regardless of discipline or academic level. The virtual teaching assistant can answer course specific questions concerning curriculum, logistics, and course policies via voice-based interface.

Several studies have focused on developing frameworks for generative AI and intelligent chatbots in higher education [45–49]. However, these frameworks have certain limitations:

- They do not fully address the operational aspects of teaching and learning in higher education and may not encompass key elements of the university syllabus, such as knowledge acquisition and skills development.
- Existing frameworks often focus on specific academic courses or support specific stakeholders involved in the educational process, such as students, teachers, administrative departments, or governing bodies.

- They tend to assess the implications of using artificial intelligence tools in higher education primarily through assessing satisfaction levels of students and educators, rather than providing algorithms for chatbot usability analysis.

Despite the availability of previous studies, a universally accepted system for the design, development, and implementation of AI chatbots in higher education is still lacking.

### 3.2. Measuring Students' Attitude toward Chatbot Services

This section investigates the factors that influence students' usage of AI chatbots and clarifies the reasons that motivate their adoption. User experience plays a pivotal role in determining whether students will continue using a smart chatbot and effectively communicate their needs, preferences, and concerns. Understanding and addressing student attitudes toward chatbot services have become a factor in ensuring the effectiveness of the learning process. Student satisfaction is a measure to determine how well educational services meet or exceed their learning expectations [48]. Over the past decade, student perception toward chatbot services have emerged as an indicator for the success of e-learning.

Analyzing the factors influencing students' perceptions toward educational chatbots is essential to comprehend how they interact with each other and how these interactions can foster positive outcomes in their collaboration. Sáiz-Manzanares et al. [49] have examined the impact of educational level and prior knowledge on the frequency of chatbot use and the satisfaction regarding the usefulness of chatbots among students in the field of health sciences. The results indicated that academic level had an impact on both the frequency of chatbot use and learning outcomes. However, prior knowledge only influenced learning outcomes. Furthermore, significant differences were observed in students' perceived satisfaction with the use of the chatbot, but not in relation to prior knowledge.

Moldt et al. [50] have investigated the attitudes of medical students toward AI and chatbots in the healthcare setting. There were high levels of agreement regarding the use of AI in administrative settings and research with health-related data. However, participants expressed concerns about inadequate data protection measures and the potential for increased workplace monitoring in the future.

Al-Emran et al. [51] have developed the chatbot acceptance–avoidance model, which uncovered the determinants of using chatbots for knowledge sharing. The PLS-SEM results supported the positive role of performance expectancy, effort expectancy, and habit, while highlighting the negative influence of perceived threats on chatbot use for knowledge sharing. However, regression analysis showed no significant effects of social influence, facilitating conditions, and hedonic motivation on chatbot use. On the other hand, the fsQCA analysis suggested that all factors might play a role in shaping the use of chatbots.

Firat [52] has investigated the implications of ChatGPT for students and universities by examining the perceptions of scholars and students. Through thematic content analysis, the author identified several key themes such as "evolution of learning and education systems" and "AI as an extension of the human brain". The study also discussed the potential benefits of AI in education and addressed the challenges and barriers associated with its integration.

The variables identified in previous studies indicate the multifaceted nature of student attitude toward AI chatbots. Managerial bodies of universities and chatbots providers can leverage these factors to ensure consistent and successful teaching and learning experience.

### 3.3. Measuring the Quality of Chatbot Services

Another aspect of educational chatbot studies involves evaluating their performance in various university courses and comparing it with results achieved by students. In a study, conducted by Talan and Kalinkara [53], the performance of ChatGPT was compared to that of undergraduate students in an anatomy course using a multiple-choice test. The results showed that ChatGPT outperformed the students in this examination.

The study of Tenakwah et al. [54] has investigated the competence and utility of Chat-GPT in answering university-level questions across a variety of academic disciplines. The responses were evaluated and graded based on the corresponding rubric for each discipline. The authors concluded that ChatGPT should be viewed as a tool to stimulate further and deeper thinking. When employed for research-focused tasks, human intelligence must be utilized to review the output for accuracy, consistency, and feasibility.

Qureshi [55] has conducted an investigation into the prospects and obstacles associated with utilizing ChatGPT as a learning and assessment tool in undergraduate computer science curriculum specifically in the context of teaching fundamental programming courses. Two groups of students were given programming challenges within Programming Contest Control ($PC^2$) environment. The results uncovered that students using ChatGPT had an advantage in terms of earned scores. However, there were inconsistencies and inaccuracies in the submitted code consequently affecting the overall performance.

Santos [56] analyzed the performance of two AI language models, ChatGPT and Bing Chat, in responding to questions related to chemistry, physics, and scientific concepts. The questions covered topics, including different types of chemical bonds. According to the author's results, one chatbot outperformed another, providing more comprehensive, detailed and accurate responses while addressing nuances and contexts.

Ibrahim et al. [57] examined the integrity of student evaluations in the era of AI chatbots. They compared the performance of ChatGPT with that of students across more than 30 university-level courses. The results indicated that ChatGPT's performance was comparable, if not superior, to that of students in numerous courses.

Although smart educational chatbots are a relatively new technology, there have been numerous studies exploring their impact on e-learning. As chatbot functionality is dynamically evolving, regular measurements and data analysis should be conducted to objectively assess its performance and progress. However, these studies often have a limited scope, focusing on specific educational level, learning platform, subject area, or individual chatbots. As a result, there is a necessity for a conceptual framework that can encompass and facilitate various learning and teaching activities across different academic disciplines, instructional modes (electronic or blended), and learning environments (formal or informal). Such a framework would provide a more comprehensive understanding of the potential of AI chatbots in education and support their effective implementation.

## 4. Framework for Chatbot-Assisted University Teaching and Learning

In this section, based on our experience from the previous semester, we present a new reference framework for incorporating AI chatbots into university educational processes.

To overcome the limitations mentioned at the end of Section 3.1., we have developed a new comprehensive reference framework (Figure 1) that offers a holistic approach to address the challenges and opportunities of integrating AI chatbots into university teaching and learning. This new framework integrates the use of intelligent chatbots (represented by green icons in Figure 1) at the course level within a blended learning environment. It consists of five stages, considering both the perspectives of instructors and students.

Stage 1. Before Course.

The instructor plans the course structure and content—description, learning outcomes, outline and schedule, required resources, assessment, and grading criteria. This stage also involves determining the overall course objectives and sequencing of topics. The student should familiarize himself/herself with the course syllabus provided by the university. In case of any questions or concerns about the course, the learner should contact their academic advisor, the instructor or teaching assistants for clarification such as course requirements, expectations, or assignments.

During this "pre-course" stage, AI chatbot can offer various options and alternatives for organizing the course. The educators should consider different factors such as curriculum requirements, course prerequisites, and students' needs. The final decision depends on instructors' expertise, experience, and understanding of the subject matter. The chatbot role

here is to serve as a supportive tool, aiding the lecturers in the decision-making process, while the instructor's expertise shapes the course.

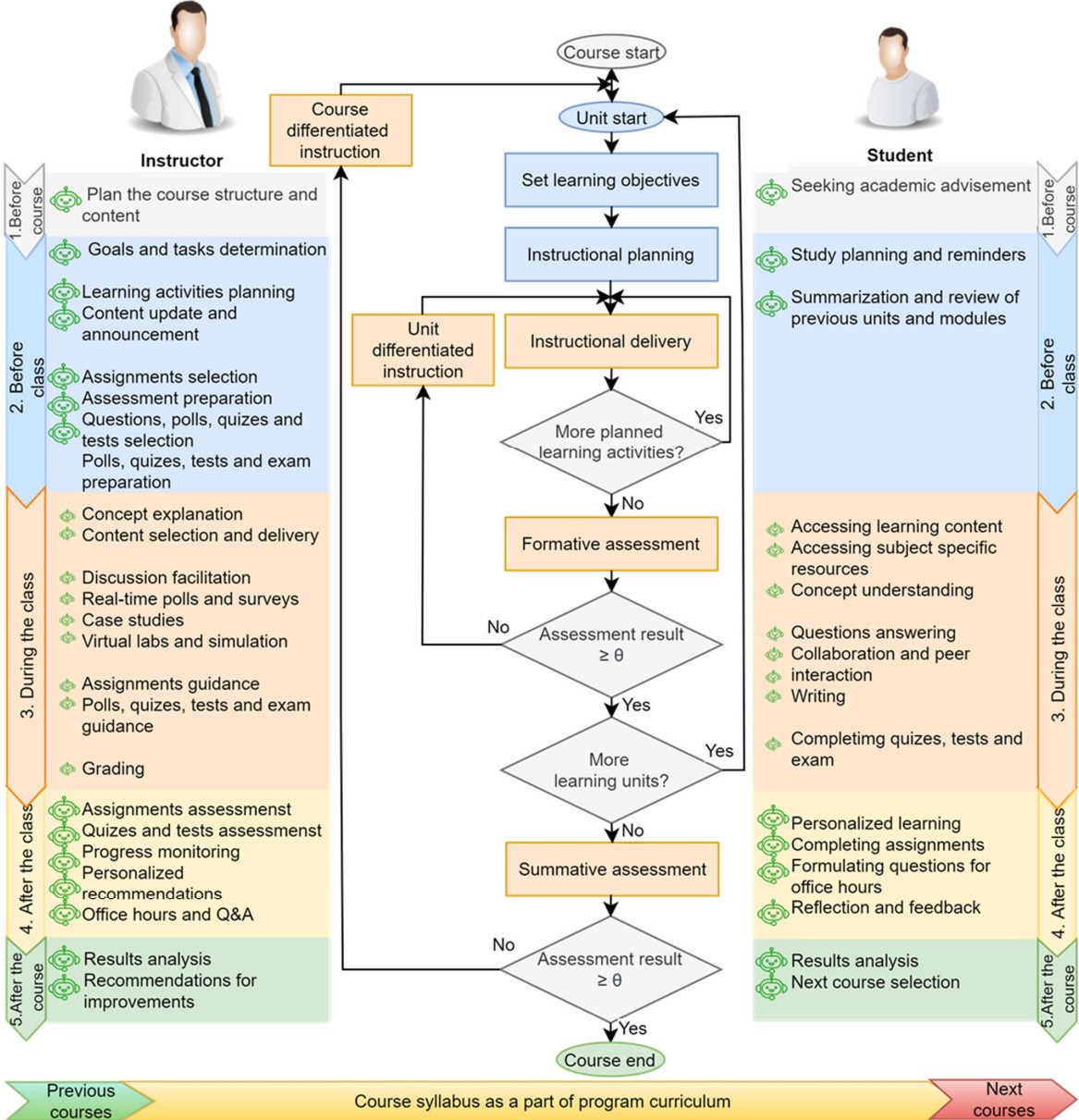

**Figure 1.** Framework for intelligent chatbot application in higher educational context. Note: The symbol θ represents the threshold (minimum passing grade) for the midterm and final exams, which is predetermined in the course syllabus and typically falls within the range of 59% to 69%.

Stage 2. Before Class.

This stage encompasses a range of activities focused on class preparation. The instructor sets learning objectives that define what the learners should know or be able to do after each course component (lesson, seminar or laboratory). The feedback instruments include polls, assignments, quizzes, tests, and mid-term exams. Before attending a class, students can take several actions to prepare themselves effectively such as reading any assigned readings, reviewing lecture notes, or searching the course material relevant to the upcoming class to be ready for class discussions or activities.

In this stage, AI chatbot can provide access to digital course materials, such as readings, lecture notes, or multimedia resources, allowing instructors and students to review them within the chat interface. The chatbot can deliver pre-class assignments or quizzes and pro-

vide immediate feedback or explanations for incorrect answers. It can also help educators and students through interactive modules or simulations related to the upcoming class topics. Here, the AI chatbot serves as a supporting instrument for university instructors. However, the instructors determine the duration and frequency of individual learning activities, select teaching and assessment methods, design assignments, and structure tests and exams.

Stage 3. During the Class.

During the class stage, the educators deliver engaging lectures, facilitate discussions, and provide explanations to ensure students' understanding. They encourage active participation, address any misconceptions, and assess student comprehension through formative assessments. Educators may also adjust the pacing or delivery of the lesson based on student feedback and comprehension levels. Meanwhile, university students engage in various tasks to maximize their learning experience. They actively participate in class discussions, ask questions to clarify concepts, take notes, and collaborate with peers on group activities. Students also practice critical thinking by analyzing data and synthesizing information presented in real-time, actively applying the knowledge to problem-solving scenarios.

In this stage, the role of chatbots is minimal as activities primarily occur in real-time and involve synchronous communication. In Figure 1, this fact is represented by the smaller size of the chatbot icons. The focus during this stage is the direct interaction and engagement between students and educators, and therefore, chatbots may not play a significant role during class sessions.

Stage 4. After the Class.

After the class, instructors may grade assignments and provide feedback to students. They may also be available within office hours for meetings or virtual consultations to address any students' questions or concerns. Students may use this time to review their class notes, consolidate their understanding of the learning material, and engage in self-study to clarify concepts covered during the class. They may work on readings, assignments, or projects related to the class topic to gain knowledge and meet deadlines. Students may also seek additional resources, participate in study teams, or communicate with instructors seek reinforcement or further instructions. Instructors and students utilize this time for independent learning, reflection, and preparation for future classes.

For instructors, AI chatbots can assist with automating administrative tasks, such as grading assignments or providing personalized feedback, allowing them to focus on individual student needs. AI chatbots can support students by providing supplementary resources, answering follow-up questions, or offering additional practice materials.

Stage 5. After the Course.

After the course's end, university instructors and students engage in various activities to wrap up their learning experience. Instructors may spend time assessing student performance, finalizing course evaluations, and reflecting on the effectiveness of their teaching methods. They may also update course materials, make adjustments based on student feedback, and prepare for next iteration of the course classes. Students may review their final grades and feedback from the instructor, reflecting on their overall performance and learning outcomes. They may clarify their strengths and areas for improvement, as well as consider how the course aligns with their academic and career goals. Students may also provide course feedback through evaluations or engage in discussions with their peers to share insights and experiences. In addition, instructors and students may explore opportunities for continuing learning or professional development related to the course topic. This could involve further research, seeking additional resources and participating in workshops or related courses to update their knowledge and skills.

The educators can employ the methodologies described in the following two sections to assess the efficiency of chatbots in a specific course.

According to the proposed framework, chatbots can enhance the teaching and learning processes by offering resources, facilitating access to materials, providing guidance on certain topics, and assisting with administrative tasks related to university courses. The

benefits for instructors are numerous. For example, chatbots can grade student works, save instructors' time, and inform students with immediate feedback on their performance. However, it is essential to emphasize that the most important decisions related to instructional design, assessment methods, and overall course management lie with the instructor. The chatbot's role is to support and complement the instructors' efforts rather than replace their expertise and decision-making capabilities. For students, chatbots provide instant responses to their questions, which reduce wait times and improve the overall student experience. Moreover, chatbots can adjust the learning experience based on each student's individual needs and preferences. They can promote critical thinking by engaging students in interactive conversations that require them to analyze, evaluate, and synthesize information to arrive at a solution or answer. This can be achieved through open-ended questions that prompt students to provide evidence to support their arguments and challenge them to consider alternative viewpoints.

By incorporating generative AI chatbots into training, fundamental didactic principles such as activity, flexibility, accessibility, consistency, and systematicity can be effectively realized. This technology enables educators to highlight the most crucial aspects of the learning material, address knowledge gaps, and facilitate the development of professional skills among students. AI chatbots offer opportunities for enhanced perception, assimilation, and application of learning content. Due to their interactive nature, AI chatbots can be effectively applied in various classroom scenarios, including systematization, summarization, and consolidation. Students can practice and repeat the learning material indefinitely, aligning with their interests and capabilities. This enables a more comprehensive and personalized learning experience [58,59].

## 5. Verification of the Proposed Chatbot-Based Teaching–Learning Framework

To verify the applicability of the proposed framework for chatbot-enhanced teaching and learning, this section employs assessment metrics to measure the satisfaction of both students and faculty members.

### 5.1. Student Survey on the Capabilities of Generative AI Chatbots—Questionnaire Design, Data Collection, and Data Analysis

To assess the usefulness of the proposed framework for students, we gathered and documented their perceptions of implementing generative AI chatbots in university learning activities. We collected students' opinions through an online survey targeting Bulgarian university students. The survey was completed on a voluntary basis and followed the principles of the Unified Theory of Acceptance and Use of Technology (UTAUT) by Venkatesh et al. [60].

The questionnaire, developed using Google Forms, comprised both close-ended (Question #1–Question #9) and open-ended (Question #10) questions. These questions were specifically designed to assess students' attitudes toward AI chatbots adoption. The survey was based on previous research on students' intentions to adopt electronic learning technologies [60–62], and it follows the format proposed by Chan and Nu [4]. The questionnaire included five main parts: introduction, demographics, frequency of use of generative AI technologies use, attitude toward smart chatbots, and future intentions. The fourth part consisted of five multiple-choice grid questions related to UTAUT concepts, with responses on a Likert-type scale from 1 to 5 (1—strongly disagree to 5—strongly agree). Question #5 (chatbot technology knowledge, six items), Question #6 (willingness to use, eight items), and Question #8 (benefits, four items) corresponded to the constructs from Venkatesh et al.'s model, enabling conditions, user intentions to use, and performance expectancy, respectively [60]. Question #4 (Frequency of use) was retrieved from Verhoeven at al. [61]. Question #7 (concerns, five items) and Question #9 (challenges, five items) utilized questions about privacy, security, and perceived risks from Chao [62]. To account for recent changes in student behavior resulting from the quick ChatGPT evolution, an open-ended question is added [63].

The questionnaire and participants' responses are available online [63]. The data regarding students' attitudes toward chatbot-assisted learning was collected between 14 May 2023, and 31 May 2023. The rules for dataset coding as well as the coded data are also accessible online. Out of the total 10 responses, 8 have been coded. The two open-text answers (related to major and opinions) have undergone additional processing.

To analyze the collected dataset and uncover hidden relationships, we utilized statistical analysis techniques (descriptive statistics, correlation analysis, and analysis of variance (ANOVA)), data mining techniques (data cleaning, data visualization, outlier detection, and cluster analysis), and multi-criteria decision-making technique. Furthermore, we employed sentiment analysis to examine students' opinions about their experience with AI chatbots.

A total of 131 respondents completed the questionnaire and 14 of them indicated that they never used a chatbot (Question #4). A duplicate check was conducted, and no identical values were found within the dataset rows. However, upon analyzing the multiple-choice grid data (Question #5 to Question #9), it was observed that dataset row #104 had a duplicate entry (#105) (Figure 2).

Figure 2 depicts the level of similarity among participants' responses, where bigger distance indicates bigger difference. The similarity level is represented by varying colors, ranging from complete similarity (0—green color) to maximum divergence (16—red color). As the dataset not contain entirely identical records, all observations will be included in the analysis. To generate the distance matrix, the R programming language's fviz_dist() function was employed.

Demographic information.

Participants in this study were 131 undergraduate and graduate students from three majors (Economics, Public Sector Management, and Business Management) in Plovdiv University Paisii Hilendarski, comprising 109 females (83.2%) and 22 males (16.8%). Additionally, 89.3% respondents have reported using generative AI at least once. Specifically, 26.0% reported rarely using it, 40.5% using it sometimes, 18.3% often using it, and 4.6% reported always using it. Table 2 shows the demographic information.

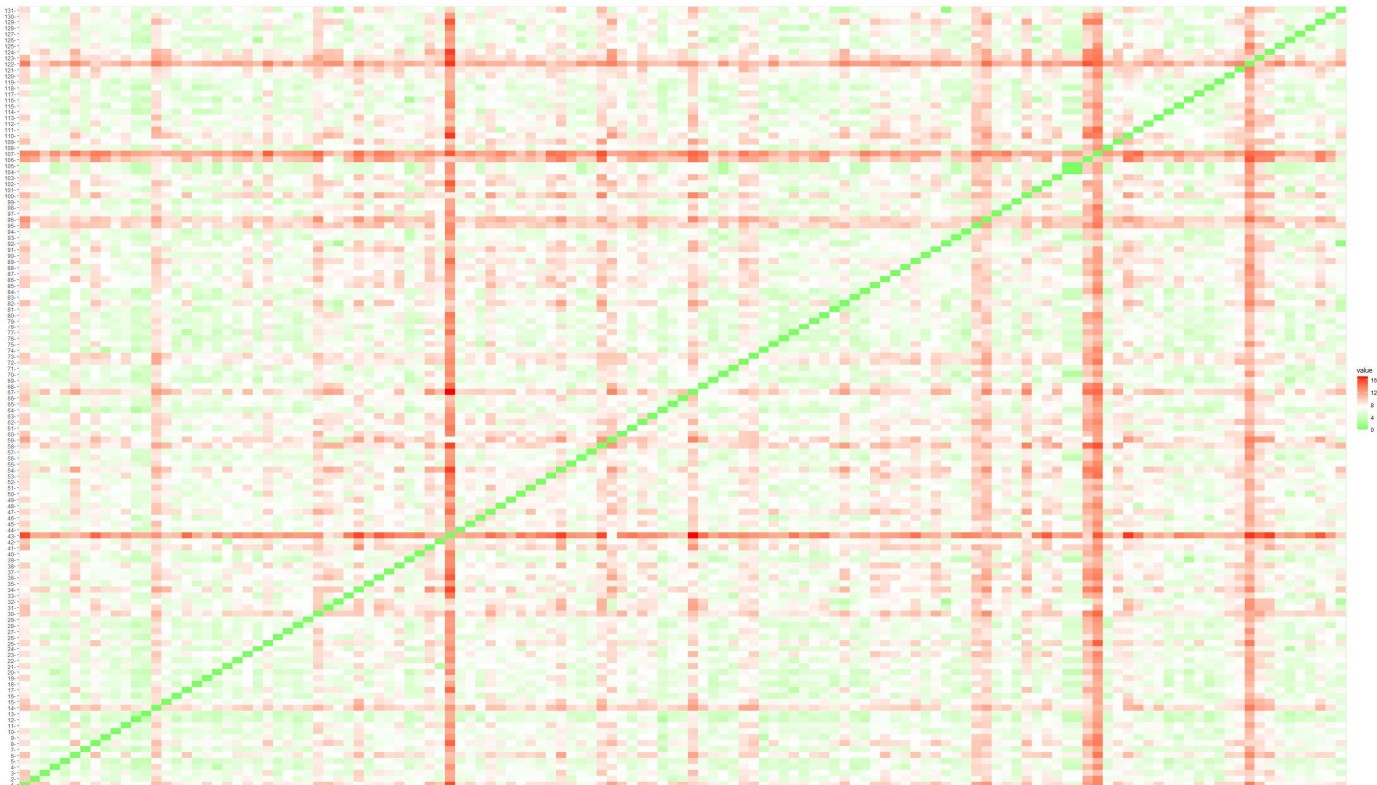

**Figure 2.** The matrix of distances (dissimilarity matrix) between students' answers.

**Table 2.** Students' characteristics in the sample (*n* = 131).

| Sample Description | | No. of Students | Percentage (%) |
|---|---|---|---|
| Question #1. Gender | Male | 22 | 16.8 |
| | Female | 109 | 83.2 |
| Question #2. Major | Economics | 103 | 78.6 |
| | Management | 17 | 13.0 |
| | Other | 11 | 8.4 |
| Question #3. Educational level | Bachelor | 128 | 97.7 |
| | Master | 3 | 2.3 |
| Question #4. Frequency of chatbot usage | Never | 14 | 10.7 |
| | Rarely | 34 | 26.0 |
| | Sometimes | 53 | 40.5 |
| | Often | 24 | 18.3 |
| | Always | 6 | 4.6 |

As shown in Table A.1. from the web Appendix [63], respondents have a good understanding of generative AI technologies with mean scores from 2.47 (item #4) to 3.40 (item #2) (Question #5). Students have the highest mean score for the statement "I understand generative AI technologies like ChatGPT can generate output that is factually inaccurate" (Mean = 3.40, SD = 0.98) [63]. The lowest mean score is for the bias and unfairness considerations (Mean = 2.47, SD = 1.02), indicating that participants may have limited awareness of the potential risks associated with chatbots biases and unfairness. Regarding respondents' agreement on whether generative AI technologies may generate output that is factually inaccurate, students who never or rarely use generative AI technologies (Mean = 3.52, SD = 0.85) are not significantly different (F = 1.037, $p > 0.05$) from students who have used them at least sometimes (Mean = 3.34, SD = 1.04).

The findings suggest that students have a positive attitude toward generative AI technologies (Question #6). They would like to integrate these technologies into their teaching and learning practices (Mean = 4.06, SD = 0.75) as well as in their careers (Mean = 3.79, SD = 0.97). Specifically, students perceive the highest values from time-saving (Mean = 4.38, SD = 0.68) and availability 24/7 (Mean = 4.26, SD = 0.78). The correlation analysis results reveal that students' perceived intention to use generative AI technologies is positively correlated with frequency of use (r = 0.330, *p*-value < 0.001). In other words, students who use them more frequently are more likely to use them in the future.

Unlike intention to use, students express some concerns about generative AI (Question #7). They convey the highest positive rating about whether these technologies hinder the development of transferable skills (Mean = 3.32, SD = 1.20), and the least positive attitude is about whether people will become overly reliant on new AI technologies (Mean = 3.14, SD = 1.33).

Question #8 and Question #9 address the benefits and challenges posed by generative AI, respectively. The mean scores of 3.39 and 3.06 imply that the majority of students seem to recognize the advantages and difficulties, associated with generative AI. This also suggests that, as a whole, students perceive the advantages of generative AI to outweigh the accompanying challenges.

Cluster analysis identifies groups of students with similar attitudes and opinions toward generative AI chatbots. To determine the optimal number of clusters for k−means clustering, we employed the Elbow method, and the result shows that the optimal number of clusters is two. As can be seen in Figure 3, when k = 2, there is no overlap between clusters. Therefore, the two clusters present a feasible solution to the problem of identifying groups of students with a similar attitude (Question #4–Question #9) toward generative AI chatbots.

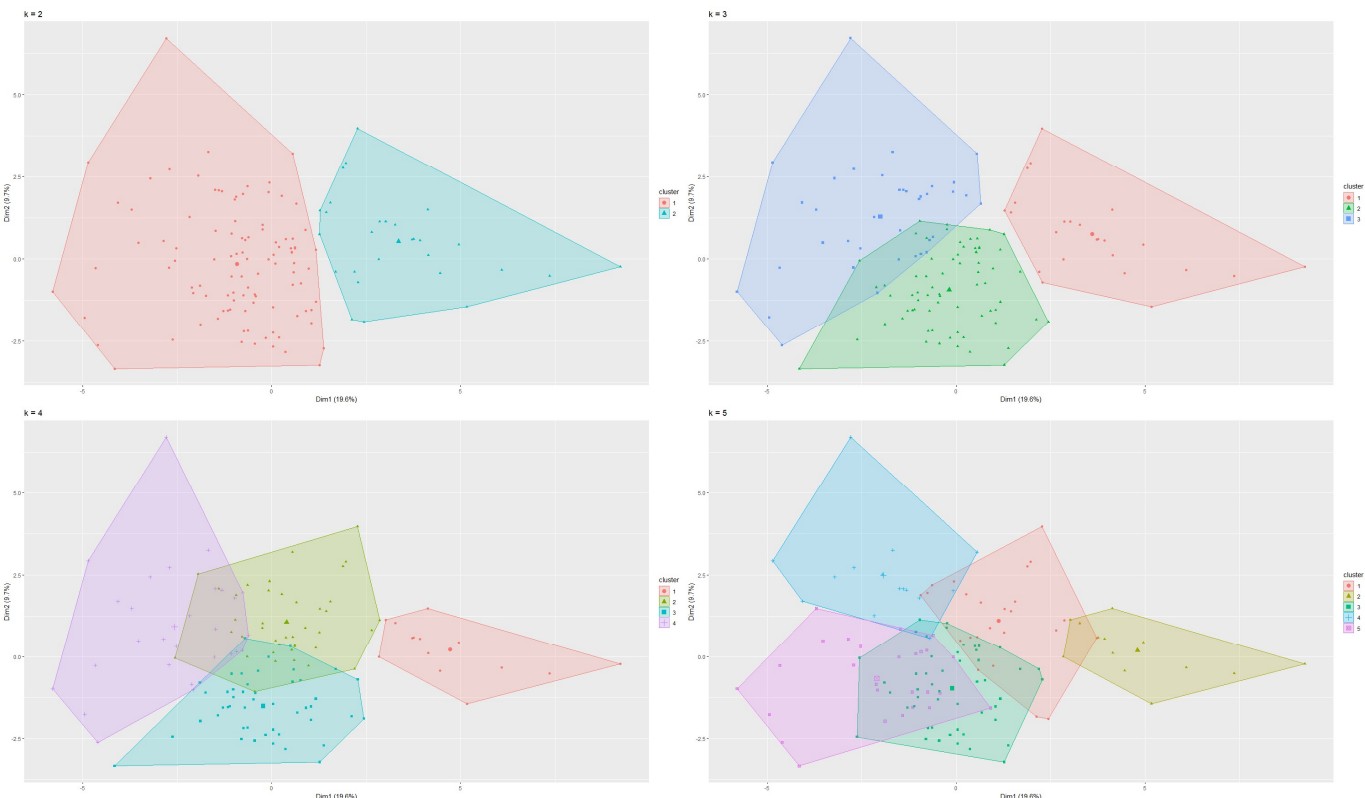

**Figure 3.** Students' clusters by k−means (k = 2, 3, 4, 5).

The first cluster consists of 103 "satisfied" students—with higher ratings on student intention to use AI chatbots (Question #6) and greater benefits from their usage (Question #8) (Table A.2). In contrast, the students from the second cluster (28 students) demonstrate lower satisfaction in generative AI technologies. Differences between students from the two clusters are insignificant in terms of knowledge (Question #5) and concerns about AI chatbot applications (Question #7). In Table A.2, the average estimates of indicators for two clusters, as well as the differences between these estimates, are depicted [63].

The last question (Question #10), which was an open-ended one, received 64 replies. After preprocessing, 54 answers remain, and answers of the type "Yes/No" were dropped. After conducting sentiment analysis, the responses were classified as follows:

- positive—27, average value 0.76;
- neutral—9, average value 0.54;
- negative—18 (actually 17, because one of the negative opinions has score 0.001), average value 0.16.

The students support the integration of chatbots in university learning as a means to accelerate knowledge acquisition and improve learning outcomes. Several advantages of integrating AI chatbots into blended learning are outlined. The students, who expressed a negative attitude primarily emphasized the importance of face-to-face teamwork for developing their soft skills. They also propose having student texts verified as non-AI-generated through sources like ChatGPT. Neutral opinions support AI chatbot implementation but point out some weaknesses in online learning. Sentiment analysis was performed using the Azure Machine Learning add-in MS Excel.

Despite the large number of studies about chatbots adoption in higher education, comparing the obtained datasets remains challenging due to variations in selected variables and analysis methodologies. Our study aligns with previous research conducted in Hong Kong universities [4], as both countries implemented AI chatbots for online learning. However, in Bulgaria, chatbot usage is more prevalent among students (89.3% and 66.7% in Bulgaria and Hong Kong, respectively). Bulgarian and Hong Kong students exhibit high

levels of their intention to use generative AI technologies (with scores ranging from 3.66 to 4.26 and from 3.61 to 4.20, respectively).

In both countries, many respondents have faced difficulties in their chatbot-assisted studies (for example, the scores for availability of factually inaccuracy are 3.40 and 4.10 in Bulgaria and Hong Kong, respectively). According to Hong Kong students, the impact on the intention to use smart chatbots of both enabling conditions (knowledge about AI technologies, r = 189) and frequency of use (r = 0.326) is similar to the impact of frequency of use (r = 0.243) according to Bulgarian students' expectations. Our findings about existing statistically significant dependencies between use behavior and intention to use AI chatbots are also consistent with those reported in [62]. The students' perceptions regarding the importance of assessing validity and accuracy and detecting falsity of chatbot statements correspond with the results of [64] and [65].

Our participants express a higher level of concern that data protection and security may be insufficiently guaranteed in comparison with those from [50], 84.7% and 33.3%, respectively. If our proposed comprehensive framework had been employed in this study, the comparison of the obtained results would have been facilitated.

The results provide insights into the way students perceive and interact with intelligent chatbots, helping instructors and developers to address adequately revealed concerns.

### 5.2. Experimental Evaluation of the Reliability of Chatbots as an Educational Tool

To assess the effectiveness of proposed framework from the educators' viewpoint, we conducted a practical experiment involving several chatbots assigned two tasks—one from Algebra and one from Financial Mathematics. The formulations of the tasks' are provided below.

Task #1.

The daily demand for a specific type of fuel at a gas station can be described by the function x(v) = a − bv^2. In this equation, v represents the price per liter of fuel, and x(v) represents the corresponding demand in liters and the symbol "^" represents the POWER function. The function x(v) exhibits the following properties:

We have the following information:

At a price of EUR 3 per liter, the demand is 810 L per day.

At a price of EUR 4 per liter, the demand is 530 L per day.

To determine the price per liter of fuel that would result in a demand of 900 L per day, we need to solve for v.

Task #2.

Victor initiates a bank account by depositing $1200. The account is subject to quarterly compounding at an annual nominal interest rate of 2.64%. From the start of the second year onwards, Victor contributes $150 to his account at the beginning of each quarter. The total amount that will accumulate in the account after 5 years needs to be calculated.

The experiment took place at the end of the month of June 2023 and each chatbot provided its solutions. The screenshots of the chatbots' responses can be found in the online Appendix [63]. Tables 3–5 display the obtained solutions and present comparisons of the chatbot's performance on these tasks. Grading criteria are derived from the cognitive process dimensions according to Bloom's modified taxonomy [66].

**Table 3.** Evaluation of chatbots' performance: Task #1 solutions.

| Chatbot | Remembering | Understanding | Applying | Analyzing | Evaluating | Creating | Total |
|---|---|---|---|---|---|---|---|
| ChatGPT | 4 | 4 | 4 | 4 | - | 4 | 20 |
| Bard | 4 | 4 | 2 | 2 | - | 1 | 13 |
| Alpaca-13B | 0 | 0 | 0 | 0 | - | 0 | 0 |
| Vicuna-13B | 4 | 4 | 2 | 1 | - | 1 | 12 |
| Vicuna-33B | 3 | 0 | 0 | 0 | - | 0 | 3 |
| ChatGPTPlus | 4 | 4 | 4 | 4 | - | 4 | 20 |
| Edge Chat | 4 | 4 | 1 | 2 | - | 4 | 15 |

**Table 4.** Evaluation of chatbots' performance: Task #2a solutions.

| Chatbot | Remembering | Understanding | Applying | Analyzing | Evaluating | Creating | Total |
|---------|-------------|---------------|----------|-----------|------------|----------|-------|
| ChatGPT | 4 | 4 | 3 | 4 | - | 4 | 19 |
| Bard | 2 | 2 | 2 | 2 | - | 1 | 9 |
| Alpaca-13B | 0 | 2 | 0 | 0 | - | 0 | 2 |
| Vicuna-13B | 0 | 0 | 0 | 0 | - | 0 | 0 |
| Vicuna-33B | 0 | 2 | 2 | 2 | - | 0 | 6 |
| ChatGPTPlus | 4 | 4 | 4 | 4 | - | 3 | 19 |
| Edge Chat | 4 | 4 | 2 | 4 | - | 4 | 18 |

**Table 5.** Evaluation of chatbots' performance: Task #2b solutions.

| Chatbot | Remembering | Understanding | Applying | Analyzing | Evaluating | Creating | Total |
|---------|-------------|---------------|----------|-----------|------------|----------|-------|
| ChatGPT | 2 | 2 | 2 | 3 | 3 | 1 | 13 |
| Bard | 1 | 1 | 1 | 1 | 0 | 0 | 4 |
| Alpaca-13B | 0 | 2 | 1 | 0 | 0 | 0 | 3 |
| Vicuna-13B | 0 | 0 | 0 | 0 | 0 | 0 | 0 |
| Vicuna-33B | 0 | 0 | 0 | 2 | 0 | 0 | 2 |
| ChatGPTPlus | 4 | 2 | 3 | 4 | 4 | 4 | 21 |
| Edge Chat | 1 | 1 | 1 | 1 | 0 | 0 | 4 |

Three lecturers who are experts in the relevant subject areas conducted the evaluations of the solutions. Each chatbot grade is determined by consensus after a collaborative discussion within the team. The rating has been performed using a 5-point scale: 0—fail, 1—below average, 2—satisfactory, 3—good, and 4—excellent. The symbol "-" means that the solution of the task does not require the corresponding level of cognitive learning from the taxonomy.

Based on the obtained results, it can be concluded that chatbots have the potential as a new educational tool, as they demonstrated successful performance to some extent. While the majority of chatbots successfully solve Task #1, the second task proves to be challenging for almost all of them. Chatbots struggle to correctly solve the second part of "trick" Task #2, as they make errors in various steps of the solution algorithm. Overall, ChatGPT Plus demonstrates the highest performance (total score of 60 out of 64 points) for all tasks, while Alpaca-13B obtains the lowest score (5 out of 64 points). This ranking aligns with expectations, as ChatGPT Plus is built upon the GPT-4 LLM, as described in Section 2.1.

Previous studies [53,54] have compared the outcomes of chatbots with those of students, indirectly examining the effectiveness of chatbots. In contrast, our approach differs as we directly evaluate the preparedness of chatbots in assisting students. Our findings are similar to those from previous research, indicating that chatbots play a valuable role in generating prompts that encourage critical thinking. However, when it comes to solving STEM tasks, human intervention is necessary to assess the accuracy, consistency, and feasibility of the chatbot's output [67].

Chatbots encounter challenges with more complex tasks and students may not always rely on their assistance. Instructors should familiarize themselves with the capabilities of AI chatbots in advance and recommend the most suitable chatbot to students for the specific topic.

## 6. Conclusions and Future Research

In this paper, we present a new conceptual framework that enables the evaluation, comparison, and prediction of the effects of conversational chatbots on university instructors' teaching and students' learning using both classical and intelligent methods for data analysis. By integrating classical and intelligent statistical methods for data analysis, we propose a comprehensive methodology for systematically evaluating students' percep-

tions and instructors' readiness concerning chatbot-assisted educational activities in the digital environment.

The new framework for AI-assisted university courses has been applied to study the effects of intelligent chatbots on electronic learning at Plovdiv University Paisii Hilendarski. The results indicate that a substantial number of students are aware of the educational potential of this emerging AI technology and have already used it. Our research also reveals that a significant majority of students (103 out of 131) demonstrated a strong intention to use AI chatbots and expressed high satisfaction with generative AI technologies. Students with a better understanding of the advantages of this technology tend to use it more frequently and express a greater intention to continue its usage. However, in terms of AI-chatbots' capabilities as an educational tool, instructors' assessments of chatbots' solutions are not impressive. In both tasks—Task #1 from Algebra and Task #2 from Financial Mathematics—only two (ChatGPT and ChatGPTPlus) out of seven chatbots received acceptable grades: 100% of the maximum possible points for Task 1, and 72.7% and 90.9% for Task #2, respectively.

Here are some recommendations for management bodies, educators, and students regarding the proposed new framework for AI chatbot-assisted university courses. Universities should develop an AI adoption strategy at the institutional level. This strategy should include plans for investing in digital innovations and upgrading the digital skills of instructors and students. Unfortunately, there is a lack of research on the level of digital skills, and one needs to first establish what the current situation is. According to a recent study by the Organization for Economic Cooperation and Development (OECD), the digital skills of Bulgarian educators are not at a high level. Digitization, however, should not be an end in itself; instead, the pedagogical goals should take the lead. Moreover, many occupations are anticipated to become obsolete, while the number of digitally enabled professions and roles is expected to rise. This shift requires a significant number of individuals to undergo career transitions throughout their lifetime. To address these evolving conditions, universities should be prepared to offer suitable lifelong learning opportunities that incorporate the use of artificial intelligence.

The study has several limitations: (1) the empirical research involved participants exclusively from Plovdiv University, limiting the generalizability of the findings; (2) some steps of the proposed framework were not fully tested, such as real-time experiments with AI chatbots during classes; (3) the analysis of the students' dataset did not include an examination of its dynamics over time, as there was a lack of previous periods data; and (4) the relationships between student and faculty performance and the degree of application of generative artificial intelligence are not explored.

Our future plans involve the following: (1) expanding the sample of participants in our survey on chatbot-assisted learning and teaching to include a broader range of individuals; (2) conducting comparative analyses between our findings and similar studies in other majors, universities, and countries, examining various attributes such as academic levels, courses, and learning activities; and (3) monitoring the developments and progress in AI-based learning technologies within the higher education environment.

**Author Contributions:** Conceptualization, G.I. and T.Y.; modeling, G.I., T.Y., M.B. and S.K.-B.; validation, G.I. and T.Y.; formal analysis, T.Y.; resources, G.I., T.Y., M.B., D.A. and S.K.-B.; writing—original draft preparation, G.I.; writing—review and editing, G.I., T.Y. and D.A.; visualization, T.Y. and S.K.-B.; supervision, G.I.; project administration, A.D.; funding acquisition, G.I., T.Y. and S.K.-B. All authors have read and agreed to the published version of the manuscript.

**Funding:** This research was funded by the National Research Program "Young Scientists and Post-Doctoral Researchers-2" (YSPDR-2), Grant No. YSPD-FESS-021, and the National Science Fund, co-founded by the European Regional Development Fund, Grant No. BG05M2OP001-1.002-0002 "Digitization of the Economy in Big Data Environment".

**Institutional Review Board Statement:** Not applicable.

**Informed Consent Statement:** Not applicable.

**Data Availability Statement:** The data stored as csv and pdf files are publicly available at https://data.mendeley.com/drafts/k8mkczry5h (accessed on 30 June 2023).

**Acknowledgments:** The authors thank the academic editor and anonymous reviewers for their insightful comments and suggestions.

**Conflicts of Interest:** The authors declare no conflict of interest.

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
