# Peer review of "Effects of Generative Chatbots in Higher Education"

_information, doi:10.3390/info14090492_

Round 1

Reviewer 1 Report

This is a very interesting work.

The authors should consider revising the "state of the art" and "related work" sections so as to organize the systems in a more effective manner. Presenting related work and state of the art systems one by one, without categorizing or grouping them based on their underlying concepts, approaches, or methodologies, does not help readers to grasp the broader picture and identify the distinguishing characteristics of different approaches more readily.

Furthermore, I would like to advise the authors to consider presenting the classification of AI chatbots' platforms and systems mentioned on page 10 in the form of a table. This addition would greatly enhance the clarity of the paper. 

Additionally, the authors should consider distinguishing the methodology they followed from the framework description in order to enhance the comprehensibility of their research. Moreover, I would like to suggest that the information regarding the participants be mentioned earlier in the text to improve the overall structure and flow of the article.

Finally, there are some formatting issues: 

- On page 7, there appears to be a highlighted text.

- On page 11, there appears to be a text strike-through

The author should carefully revise the article so as to eliminate various mistakes. For example:

- ...of the paper is the development of new conceptual framework...

- This reference framework allows systematically assessment of students’ ...

Author Response

Dear respected Reviewer,

Thank you very much for providing us with the opportunity to revise our manuscript titled “Effects of Generative Chatbots in Higher Education”. Here, we have included the list of changes and responses to the reviewer‘s comments.

In the revised manuscript, we have marked corrections in red, highlighted them and uploaded the paper for your further consideration.

Answers to Reviewer #1 remarks:

Thank you for your detailed review and valuable comments.

Remark 1.

This is a very interesting work.

Answer to Remark 1.

Thank you for your accurate review and precise evaluation!

Remark 2.

The authors should consider revising the "state of the art" and "related work" sections so as to organize the systems in a more effective manner. Presenting related work and state of the art systems one by one, without categorizing or grouping them based on their underlying concepts, approaches, or methodologies, does not help readers to grasp the broader picture and identify the distinguishing characteristics of different approaches more readily.

Answer to Remark 2.

Thank you for your helpful remark. In “2. State of the Art Review of Intelligent Chatbot Models, Platforms and Systems” section we have presented several classifications of the most widely used LLMs depending on different criteria (supported platforms, access type, modalities, licensing model, etc.). We have revised “3. Related Work” section according to your kind suggestion. We have included grouping of existing approaches described in previous similar studies.

Remark 3.

Furthermore, I would like to advise the authors to consider presenting the classification of AI chatbots' platforms and systems mentioned on page 10 in the form of a table. This addition would greatly enhance the clarity of the paper. 

Answer to Remark 3.

Thank you for your valuable advice. Generative chatbots have experienced rapid evolution in recent months. Additionally, different chatbots have highly distinct characteristics. For example, some of them are finished commercial products (ChatGPT-4), while others have limited access (restricted number of chat messages per session and per day) (Bing Chat). Some chatbots have limited use or are still in a testing period (ERNIE Bot, PanGu-Bot, Yandex YaLM chatbot). In addition, the functionality of chatbots allows them to be customized for specific application areas, making them more refined and effective. Given the rapid evolution of chatbots and our desire not to rely on marketing data, we have chosen not to include a chatbot comparison table at this time. Instead, the summary of the presented chatbots has been provided in a text form, based on eight criteria (functionality, language support, Internet connectivity, multi-modality, price model, ease of use, set up complexity, and use cases) on p. 11. We have preferred this format over a tabular one due to potential challenges such as missing up-to-date data, restricted access to certain characteristics, or the absence of standardized legal rules in the EU at the moment. Additionally, we have already prepared a dedicated subsection (“2.1. Large Language Models for NLP and Their Comparison”), focused on LLMs. As LLMs significantly predetermine the characteristics of the respective chatbots, we have conducted a detailed comparison of their features, and the obtained results have been presented in a tabular form (Table 1). 

Remark 4.

Additionally, the authors should consider distinguishing the methodology they followed from the framework description in order to enhance the comprehensibility of their research. Moreover, I would like to suggest that the information regarding the participants be mentioned earlier in the text to improve the overall structure and flow of the article.

Answer to Remark 4.

Thank you for your insightful comment.

Answer to Remark 4а.

According to your suggestion, the structure of section “4. Research Methodology” has been transformed into two sections as follows: “4. Framework for Chatbot-assisted University Teaching and Learning” and “5. Verification of Proposed Chatbot-based Teaching-Learning Framework”.  The fifth section includes two verification parts: “5.1. Student Survey on the Capabilities of Generative AI Chatbots – Questionnaire Design, Data Collection and Data Analysis” and “5.2. Experimental Evaluation of the Reliability of Chatbots as an Educational Tool”, corresponding to the previous subsections “4.2. Questionnaire Design and Data Collection” and “4.3. Experimental Setup”.  

Answer to Remark 4b.

As per your suggestion, a clarification has been added for participants in the experimental study. Each subsection of the verification section “5. Verification of Proposed Chatbot-based Teaching-Learning Framework” begins with a brief description of the methods used and presents the results obtained. Following these formats, information about participants (students and instructors) is briefly mentioned at the beginning of the two subsections.

Remark 5.

Finally, there are some formatting issues: 

- On page 7, there appears to be a highlighted text.

- On page 11, there appears to be a text strike-through.

Answer to Remark 5.

Thank you for your feedback. The text’s fragments have been reformatted.

Remark 6. Comments on the Quality of English Language

The author should carefully revise the article so as to eliminate various mistakes. For example:

- ...of the paper is the development of new conceptual framework...

- This reference framework allows systematically assessment of students’ ...

Answer to Remark 6.

Thank you for your remark. The manuscript text has been checked and edited regarding English grammar and spelling.

Reviewer 2 Report

Generative Chatbots in Higher Education: A Comprehensive Framework for Transformation

Learning technologies often fail to meet universities' requirements for learner engagement, lacking interactivity and real-time feedback. This poses challenges in providing personalized learning experiences and increases instructors' workload for maintenance and updates. However, generative artificial intelligence (AI) chatbots offer a promising solution. This study explores and compares existing educational chatbots, proposing a novel theoretical framework for blended learning. By integrating intelligent chatbots, students can interact online, while instructors can create and manage courses using generative AI tools. This framework brings several advantages: 1) a deep understanding of AI chatbots' transformative potential in education and effective implementation, 2) a holistic methodology to enhance the overall educational experience, and 3) a unified application of intelligent chatbots in university teaching-learning activities.

The topic is extremely current and exciting. This type of research effort in Chatbot is very helpful to educators and students. Overall, it is well-designed and implemented. It covers a comprehensive introduction and review of the related literature. The methodology is clearly stated, and the results and conclusions are adequately presented. However, it appears that while Figures 2 and 3 demonstrate artistic visualization, they do not clearly represent the data analysis. Additionally, the texts in Figure 3 are blurry and not readable. Please rework these figures for clarity and reinsert them into the article.

Needs minor editing.

Author Response

Dear respected Reviewer,

Thank you very much for providing us with the opportunity to revise our manuscript titled “Effects of Generative Chatbots in Higher Education”. Here, we have included the list of changes and responses to the reviewer's comments.

In the revised manuscript, we have marked corrections in red, highlighted them and uploaded the paper for your further consideration.

Answers to Reviewer #2 remarks:

Thank you for your detailed review and valuable comments.

Remark 1.

Learning technologies often fail to meet universities' requirements for learner engagement, lacking interactivity and real-time feedback. This poses challenges in providing personalized learning experiences and increases instructors' workload for maintenance and updates. However, generative artificial intelligence (AI) chatbots offer a promising solution. This study explores and compares existing educational chatbots, proposing a novel theoretical framework for blended learning. By integrating intelligent chatbots, students can interact online, while instructors can create and manage courses using generative AI tools. This framework brings several advantages: 1) a deep understanding of AI chatbots' transformative potential in education and effective implementation, 2) a holistic methodology to enhance the overall educational experience, and 3) a unified application of intelligent chatbots in university teaching-learning activities.

Answer to Remark 1.

Thank you for your valuable remarks and positive attitude!

Remark 2.

The topic is extremely current and exciting. This type of research effort in Chatbot is very helpful to educators and students. Overall, it is well-designed and implemented. It covers a comprehensive introduction and review of the related literature. The methodology is clearly stated, and the results and conclusions are adequately presented.

Answer to Remark 2.

Thank you very much for this comment!

Remark 3.

However, it appears that while Figures 2 and 3 demonstrate artistic visualization, they do not clearly represent the data analysis. Additionally, the texts in Figure 3 are blurry and not readable. Please rework these figures for clarity and reinsert them into the article.

Answer to Remark 3.

Thank you for your useful remarks.

Answer to Remark 3a.

Figure 2 visualizes the differences between students’ attitudes toward chatbots. In this case, we employ a predefined R programming language function fviz_dist() from factoextra package. This function is a widely used statistical instrument for visualizing of distances between elements using three different colors. In our case, we select green for small distances or high similarity; yellow for intermediate distances or moderate levels of dissimilarity and red for larger distances or high dissimilarity between answers/participants. As this figure is comprised of 131x131 squares, it becomes challenging to view the labels of its rows and columns elements due to their high density and small size.

Figure 3 shows the results of the R function kmeans() from the stats package, using the k-means algorithm for k = 2 to 5, with a feasible solution for k = 2. The students' belonging to the two clusters is further discussed. Here, due to the high resolution of the PNG file, blurring may also occur.

Answer to Remark 3b.

In order to improve the readability, Figure 2 and Figure 3 have been reformatted as per the authors’ instructions of Information journal.

Remark 4. Comments on the Quality of English Language

Needs minor editing.

Answer to Remark 4.

Thank you for your remark. The manuscript text has been checked and edited regarding English grammar and spelling.

Reviewer 3 Report

This paper analyzes AI based chatbots as a framework to include in a classroom setting, to increase interactivity and facilitate interaction between the instructor and students. 

The introduction is very informative, but I found it to be very long for the scope of the paper. The background knowledge is almost half of the paper presented. 

The framework is very theoretical and I am not sure it could be labeled as framework. It is based on several tasks that Chatbots can perform before, within, or after the class. 

The survey part had a large number of students. It was not gender balanced and there were no instructors involved (only students). Figure 2 is unclear and must be replaced (Figure 3 as well, the legend is unreadable)

The experiment part must be expanded and explained more. I wasn't sure what was being evaluated in the experiment and what the tables at the end of the experiments mean. 

I am not a fan of online appendices, as some people might be still reading a hard copy of the article. 

The conclusion and the motive of the paper is unclear. It seems to be tackling several tasks at once without a clear conclusion. 

Additional notes: some acronyms are not defined, typo in stage 1  on p15 (yourself) . 

Author Response

Dear respected Reviewer,

Thank you very much for providing us with the opportunity to revise our manuscript titled “Effects of Generative Chatbots in Higher Education”. Here, we have included the list of changes and responses to the reviewer's comments.

In the revised manuscript, we have marked corrections in red, highlighted them and uploaded the paper for your further consideration.

Answers to Reviewer #3 remarks:

Thank you for your detailed review and valuable comments.

Remark 1.

This paper analyzes AI based chatbots as a framework to include in a classroom setting, to increase interactivity and facilitate interaction between the instructor and students. 

Answer to Remark 1.

Thank you for your remark.

Remark 2.

The introduction is very informative, but I found it to be very long for the scope of the paper. The background knowledge is almost half of the paper presented. 

Answer to Remark 2.

Thank you for your valuable remark. The introduction spans 1 ½ pages, which adheres to the recommended length for journal article introductions. The state-of-the-art section occupies 4 ½ pages out of 27, and in our opinion, it may not appear excessively long due to its coverage of three relatively new research topics: large language models, chatbots built upon these models, and educational chatbots based on generative AI. However, following your suggestion, the section has been edited and some fragments have been removed.

Remark 3.

The framework is very theoretical and I am not sure it could be labeled as framework. It is based on several tasks that Chatbots can perform before, within, or after the class. 

Answer to Remark 3.

Thank you for your remark. We agree that the proposed framework is conceptual (theoretical).

Remark 4.

The survey part had a large number of students. It was not gender balanced and there were no instructors involved (only students).

Answer to Remark 4.

Thank you for your remark.

Answer to Remark 4a.

The dataset is not gender-balanced, mainly due to the prevalence of female respondents in respondents’ majors. However, this fact does not affect the statistical significance of the obtained results, as we are not examining the influence of the gender factor.

Answer to Remark 4b.

You are correct. Our primary focus is on gathering students’ opinions regarding educational chatbots (p. 2, Task #2). As for educators, the benefits of implementing chatbots are directly related to their experience using them. Therefore, instead of conducting a survey among educators, we propose a method for evaluating the effectiveness of AI chatbots in handling university learning tasks (p. 2, Task #5).

Remark 5.

Figure 2 is unclear and must be replaced (Figure 3 as well, the legend is unreadable)

Answer to Remark 5.

Thank you for your remark. In order to improve the readability, Figure 2 and Figure 3 have been reformatted as per the authors’ instructions of Information journal.

Remark 6.

The experiment part must be expanded and explained more. I wasn't sure what was being evaluated in the experiment and what the tables  at the end of the experiments mean. 

Answer to Remark 6.

Thank you for your valuable remark.

Answer to Remark 6а.

According to your suggestion, the structure of section “4. Research Methodology” has been transformed into two sections as follows: “4. Framework for Chatbot-assisted University Teaching and Learning” and “5. Verification of the Proposed Chatbot-based Teaching-Learning Framework”.  The fifth section includes two verification parts: “5.1. Student Survey on the Capabilities of Generative AI Chatbots – Questionnaire Design, Data Collection and Data Analysis” and “5.2. Experimental Evaluation of the Reliability of Chatbots as an Educational Tool”, corresponding to the previous subsections “4.2. Questionnaire Design and Data Collection” and “4.3. Experimental Setup”. The experimental part has been edited.

Answer to Remark 6b.

The assessment of the solutions for the two tasks provided by the eight chatbots was conducted by a team of three instructors. Tables 3 - 5 display the obtained scores for six criteria, along with the total chatbot score, using a five-point scale.

Remark 7.

I am not a fan of online appendices, as some people might be still reading a hard copy of the article. 

Answer to Remark 7.

Thank you for the remark. In our opinion, the manuscript comprehensively presents the problem, the proposed solution and the obtained results. However, some readers may also request access to the primary data and intermediate results. For this reason, we have created a web appendix.

Remark 8.

The conclusion and the motive of the paper is unclear. It seems to be tackling several tasks at once without a clear conclusion. 

Answer to Remark 8.

Thank you for your valuable remark. The conclusions section has been rewritten.

Remark 9.

Additional notes: some acronyms are not defined, typo in stage 1  on p15 (yourself) . 

Answer to Remark 9.

Thank you for your useful remark. The manuscript has been edited and the abbreviations have been defined before their first usage. 

Round 2

Reviewer 1 Report

The authors have addressed some of my comments.

As I mentioned in my previous review, presenting related work and state of the art systems one by one, without categorizing or grouping them based on their underlying concepts, approaches, or methodologies, does not help readers to grasp the broader picture and identify the distinguishing characteristics of different approaches more readily. Additionally, this makes the introductory sessions very long and thus very tiring.

Furthermore, the way that they present the the classification of AI chatbots' is not appropriate. They divide the text into small paragraphs under the sentence "Depending on their main features, the above-mentioned AI chatbots’ platforms and systems can be classified based on several criteria." If a table is not appropriate, then they should consider a bulleted list.

No comments

Author Response

Dear respected reviewer,

Thank you for providing us with a second opportunity to revise our manuscript titled “Effects of Generative Chatbots in Higher Education”. We hope the corrections address your remarks appropriately.

In this new version of the manuscript, all revisions are in blue color and highlighted in blue for better visibility and clarity.

Answers to Reviewer 1 remarks

The authors have addressed some of my comments.

Remark 1.

As I mentioned in my previous review, presenting related work and state of the art systems one by one, without categorizing or grouping them based on their underlying concepts, approaches, or methodologies, does not help readers to grasp the broader picture and identify the distinguishing characteristics of different approaches more readily. Additionally, this makes the introductory sessions very long and thus very tiring.

Answer to Remark 1.

Thank you for your valuable remarks.

Regarding the one-by-one presentation:

1) In the introduction part, the subsection “2.1. Large Language Models for NLP and Their Comparison“ presents an overview of 3 LLMs for three widely spoken languages. This review outlines the current state of research in the field of generative AI. We believe that this manuscript part is concise, informative and useful for readers, as this scientific area has been developing extremely rapidly in the recent months. If the language models in this subsection are not individually (one by one) presented but only summarized, it may potentially create difficulty in comprehending the remaining sections. Therefore, it is essential to provide a detailed presentation of each language model to ensure clarity and coherence throughout the manuscript.

This approach is also followed in the subsequent subsections, namely “2.2. Intelligent Chatbots and Their Comparison” and “2.3. Educational AI Chatbots”, where individual presentations of chatbots are provided, followed by classifications based on key features. In our opinion, this systematic, detailed and comprehensive approach is beneficial, particularly for readers unfamiliar with the specific topic of the manuscript.

2) The instructions for authors of the journal Information do not impose any limitations on the one-by-one presentation of previous similar research. Accordingly, section “3. Related Work” outlines the current state of the research field by citing key publications one by one and by groups. This approach allows for a comprehensive and in-depth analysis, aligning well with the journal’s guidelines and facilitating a thorough understanding of the research landscape for readers.

3) Additionally, we are convinced that transitioning to a “pure” summary format for sections “2. State of the Art Review of Intelligent Chatbot Models, Platforms and Systems” and “3. Related Work” could potentially compromise the defense of our scientific contributions. The current detailed presentation is essential to highlight the unique aspects of our research and its relevance in the context of existing literature.

Regarding your concern about the structure of the manuscript being unbalanced:

  1. The topicality of the subject requires a more detailed presentation of the research landscape. That is why the manuscript is of the Research article type, however, it also includes some features of a Review article.
  2. The last manuscript’s version can be divided into two parts: an introductory part (14 pages) and a main part (13 pages) out of 27 pages total length. Additionally, an online appendix (13 pages) refers to the main part. Considering this fact that the manuscript's length is not too large (less than 30 p.) and the manuscript structure in terms of proportion between introductory/main part is acceptable: 14/(27+13)=14/40 » 0.35.

Based on these two arguments, the current structure of the manuscript is justified, considering the combination of Review and Research Article elements, and the overall proportion of the sections seems appropriate, taking into account the length of the online appendix as well.

Remark 2.

Furthermore, the way that they present the the classification of AI chatbots' is not appropriate. They divide the text into small paragraphs under the sentence "Depending on their main features, the above-mentioned AI chatbots’ platforms and systems can be classified based on several criteria." If a table is not appropriate, then they should consider a bulleted list.

Answer to Remark 2.

Thank you for the useful suggestion. The criteria for chatbot classification have been reformatted as a bulleted list.

Reviewer 3 Report

I thank the authors for the modifications. 

there are still comments that need some work:

- the figures are still not clear. 

-  Section 5.2 is still not clear

- When I mention that the introduction is too long, I didn't mean specifically the Introduction art. I meant all the parts leading to section 4. 

- How is this a framework? this is still unanswered. 

thank you 

Author Response

Dear respected reviewer,

Thank you for providing us with a second opportunity to revise our manuscript titled “Effects of Generative Chatbots in Higher Education”. We hope the corrections address your remarks appropriately.

In this new version of the manuscript, all revisions are in blue color and highlighted in blue for better visibility and clarity.

Answers to Reviewer 2 remarks

I thank the authors for the modifications. 

Thank you for your positive attitude!

there are still comments that need some work:

Remark 1.

- the figures are still not clear. 

Answer to Remark 1.

Thank you for your remark. In the new version of our manuscript, Figure 2 and Figure 3 have been reformatted according to the Author’s instructions of the Information journal. Due to the journal’s requirement for a sufficiently high resolution (at least 1000 pixels height/width), the paper-based version may appear blurred. In the online version, the quality of these images will be high.

Remark 2.

-  Section 5.2 is still not clear

Answer to Remark 2.

In subsection “5.2. Experimental Evaluation of the Reliability of Chatbots as an Educational Tool”, we have introduced a new method for evaluating chatbot performance. This method involves utilizing dimensions of the cognitive process based on Bloom’s modified taxonomy as assessment criteria. The adoption of this evaluation method represents a contribution to the research in the scientific area addressed in our manuscript.

In response to your remark, we have included additional explanations about the conducted experiment and the evaluations of chatbot performance.

Remark 3.

- When I mention that the introduction is too long, I didn't mean specifically the Introduction art. I meant all the parts leading to section 4.

Answer to Remark 3.

In the introduction part, the subsection “2.1. Large Language Models for NLP and Their Comparison“ presents an overview of 3 LLMs for three widely spoken languages. This review outlines the current state of research in the field of generative AI. We believe that this manuscript part is concise, informative and useful for readers, as this scientific area has been developing extremely rapidly in the recent months. If the language models in this subsection are not individually presented but only summarized, it may potentially create difficulty in comprehending the remaining sections. Therefore, it is essential to provide a detailed presentation of each language model to ensure clarity and coherence throughout the manuscript.

This approach is also followed in the subsequent subsections, namely “2.2. Intelligent Chatbots and Their Comparison” and “2.3. Educational AI Chatbots”, where individual presentations of chatbots are provided, followed by classifications based on key features. In our opinion, this systematic, detailed and comprehensive approach is beneficial, particularly for readers unfamiliar with the specific topic of the manuscript.

Additionally, we are convinced that transitioning to a shorter format for sections “2. State of the Art Review of Intelligent Chatbot Models, Platforms and Systems” and “3. Related Work” could potentially compromise the defense of our scientific contributions. The current detailed presentation is essential to highlight the unique aspects of our research and its relevance in the context of existing literature.

Regarding your concern about the structure of the manuscript being unbalanced:

  1. The topicality of the subject requires a more detailed presentation of the research landscape. That is why the manuscript is of the Research article type, however, it also includes some features of a Review article.
  2. The last manuscript’s version can be divided into two parts: an introductory part (14 pages) and a main part (13 pages) out of 27 pages total length. Additionally, an online appendix (13 pages) refers to the main part. Considering this fact that the manuscript's length is not too large (less than 30 p.) and the manuscript structure in terms of proportion between introductory/main part is acceptable: 14/(27+13)=14/40 ~ 0.35.

Based on these two arguments, the current structure of the manuscript is justified, considering the combination of Review and Research Article elements, and the overall proportion of the sections seems appropriate, taking into account the length of the online appendix as well.

Remark 4.

- How is this a framework? this is still unanswered. 

Answer to Remark 4.

The purpose of our research is to study the effects of generative chatbots in higher education. Given that the implementation of generative AI chatbots in education is still at an early stage, we believe that a comprehensive exploration of the diverse capabilities of this new educational tool is essential.

The scheme presented on page 17 is not a conceptual framework of our research, but rather a methodological framework (model, algorithm or guidance). It outlines the role of chatbots in the learning process in higher education and comprises of two mirror components. These components visually represent the individual pedagogical activities (instructor side) and learning activities (student side) unit by unit from the course syllabus. The framework sequentially visualizes these activities and highlights the potential for their automation through AI chatbots. Some parts of this framework have been validated in the subsections “5.1. Student Survey on the Capabilities of Generative AI Chatbots – Questionnaire Design, Data Collection, and Data Analysis” and “5.2. Experimental Evaluation of the Reliability of Chatbots as an Educational Tool”.

We assert that the scheme depicted on page 17 can be termed a framework because:

- The provided flowchart systematizes all activities in the educational process related to AI chatbots, offering a structured representation.

- It contributes to expanding existing knowledge by shedding light on previously unexplored areas of AI chatbot applications, especially in higher education.

- Its holistic structure may facilitate other researchers in building upon and extending studies in this field in the future.

While the scheme is not a conceptual framework in the traditional sense, it serves as a methodological framework that outlines the utilization and potential of AI chatbots in higher education.

Round 3

Reviewer 1 Report

The authors have addressed some of my comments.

Author Response

Dear respected reviewer,

Thank you for your positive attitude.

Reviewer 3 Report

I thank the author for their second modification of the paper. 

If you look at the graphs that are not clear,  the legend and the axis labels are completely unreadable, the graph units is also not clear. 

In section 5.2, details are still missing such as were all chat bots used in the experiment for evaluation. How were task 1 and 2 performed? how were the numbers reached? for e.g. For remembering, why did chatgpt receive 4 while vicuna-33B receive 3? what did the instructors exactly evaluate? 

I selected major revision again, but I will leave it to the editors to see if this falls under minor revisions. 

Author Response

Dear respected reviewer,

Thank you for providing us with a new opportunity to revise our manuscript titled “Effects of Generative Chatbots in Higher Education”. We hope the corrections address your remarks appropriately.

In this new version of the manuscript, all revisions are highlighted in green for better visibility and clarity.

I thank the author for their second modification of the paper. 

Answers to Reviewer 3 remarks

Remark 1.

If you look at the graphs that are not clear,  the legend and the axis labels are completely unreadable, the graph units is also not clear. 

Answer to Remark 1.

Thank you for your remark. Due to the axes x and y and the clusters containing 131 labels (one per each observation), the paper-based versions of Figure 2 and Figure 3 may appear blurred. In the online version, the quality of these images will be high. Additionally, in the manuscript, the legend has been described in detail. Its online quality will also be high.

Remark 2a.

In section 5.2, details are still missing such as were all chat bots used in the experiment for evaluation.

Answer to Remark 2a. 

In subsection “5.2. Experimental Evaluation of the Reliability of Chatbots as an Educational Tool”, we utilized seven out of 12 chatbots described in subsection “2.2. Intelligent Chatbots and Their Comparison”. The preferred chatbots selected for our experiments are as follows: ChatGPT (#3), Bard (#4), Alpaca (#9), Vicuna (#10 in two versions), ChatGPTPlus (#11), and EdgeChat (#12). The remaining presented chatbots did not participate in our experiments for various reasons. Specifically, IBM Watson Assistant (#1) and Amazon Lex (#2) were not included due to their specific setup requirements and configurations. PanGu-Bot (#4) and ChatGLM (#8) did not participate due to registration requirements. Yandex chatbot (#6) and BLOOMChat (#7) did not have publicly available software versions during the preparation of our manuscript.

Remark 2b.

How were task 1 and 2 performed?

Answer to Remark 2b. 

The chatbots were assigned the two tasks, which were formulated exactly as described in our manuscript (subsection “5.2. Experimental Evaluation of the Reliability of Chatbots as an Educational Tool”). The obtained chatbot solutions are available online in our Mendeley dataset (https://data.mendeley.com/datasets/k8mkczry5h/1).

Remark 2c.

how were the numbers reached? for e.g. For remembering, why did chatgpt receive 4 while vicuna-33B receive 3? what did the instructors exactly evaluate? 

Answer to Remark 2c. 

In the same subsection, the instructors-participants in our experiment employ a specific method to evaluate chatbot performance. This evaluation method utilizes dimensions of the cognitive process based on Bloom’s modified taxonomy as assessment criteria. Using these criteria, instructors assess the chatbot’s solutions in a manner similar to how they would evaluate students’ solutions on the same tasks. The obtained results are presented in Table 3 to Table 5.

Remark 3.

I selected major revision again, but I will leave it to the editors to see if this falls under minor revisions.

Answer to Remark 3.

Thank you.